# Uncovering the Representation of Spiking Neural Networks Trained with Surrogate Gradient

**Yuhang Li**                                                          *yuhang.li@yale.edu*
*Yale University*

**Youngeun Kim**                                                       *youngeun.kim@yale.edu*
*Yale University*

**Hyoungseob Park**                                                    *hyoungseob.park@yale.edu*
*Yale University*

**Priyadarshini Panda**                                               *priya.panda@yale.edu*
*Yale University*

**Reviewed on OpenReview:** *https://openreview.net/forum?id=s9efQF3QW1*

## Abstract

Spiking Neural Networks (SNNs) are recognized as the candidate for the next-generation neural networks due to their bio-plausibility and energy efficiency. Recently, researchers have demonstrated that SNNs are able to achieve nearly state-of-the-art performance in image recognition tasks using surrogate gradient training. However, some essential questions exist pertaining to SNNs that are little studied: *Do SNNs trained with surrogate gradient learn different representations from traditional Artificial Neural Networks (ANNs)? Does the time dimension in SNNs provide unique representation power?* In this paper, we aim to answer these questions by conducting a representation similarity analysis between SNNs and ANNs using Centered Kernel Alignment (CKA). We start by analyzing the spatial dimension of the networks, including both the width and the depth. Furthermore, our analysis of residual connections shows that SNNs learn a periodic pattern, which rectifies the representations in SNNs to be ANN-like. We additionally investigate the effect of the time dimension on SNN representation, finding that deeper layers encourage more dynamics along the time dimension. We also investigate the impact of input data such as event-stream data and adversarial attacks. Our work uncovers a host of new findings of representations in SNNs. We hope this work will inspire future research to fully comprehend the representation power of SNNs. Code is released at https://github.com/Intelligent-Computing-Lab-Yale/SNNCKA.

## 1 Introduction

Lately, Spiking Neural Networks (SNNs) (Tavanaei et al., 2019; Roy et al., 2019; Deng et al., 2020; Panda et al., 2020; Christensen et al., 2022) have received increasing attention thanks to their biology-inspired neuron activation and efficient neuromorphic computation. SNNs process information with binary spike representation and therefore avoid the need for multiplication operations during inference. Neuromorphic hardware such as TrueNorth (Akopyan et al., 2015) and Loihi (Davies et al., 2018) demonstrate that SNNs can save energy by orders of magnitude compared to Artificial Neural Networks (ANNs).

Although SNNs can bring enormous energy efficiency in inference, training SNNs is notoriously hard because of their spiking activation function. This function returns a zero-but-all gradient (*i.e.,* Dirac delta function) and thus makes gradient-based optimization infeasible. To circumvent this problem, various training techniques have been proposed. For example, spike-timing-dependent plasticity (STDP) (Rao & Sejnowski, 2001)

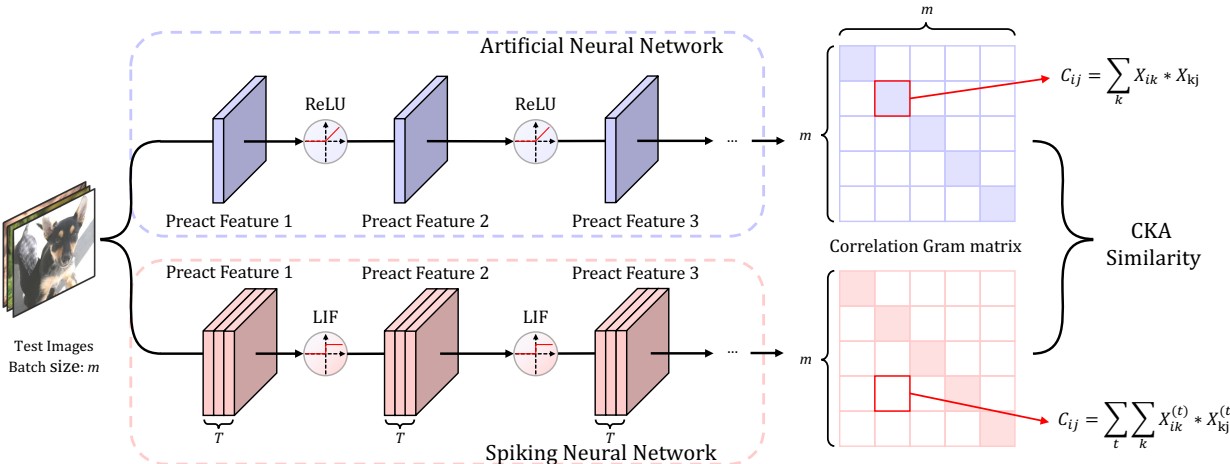

Figure 1: **The representation similarity analysis workflow.** The test images are fed into both ANN and SNN, then we record the intermediate feature for computing the correlation matrix, which is used for inferring the CKA similarity (Kornblith et al., 2019).

either strengthens or weakens the synaptic weight based on the firing time; time-to-first-spike (Mostafa, 2017) encodes the information into the time of spike arrival to get a closed-form solution of the gradients. However, these two methods are restricted to small-scale tasks and datasets. Surrogate gradient technique (Bengio et al., 2013; Bender et al., 2018; Wu et al., 2018; Bellec et al., 2018; Kim et al., 2022b; Li et al., 2022), on the other hand, can achieve the best task performance by applying an alternate function during backprop-agation. Combined with surrogate gradient, SNNs can be optimized by Backpropagation Through Time (BPTT) (Neftci et al., 2019) algorithm, outperforming other learning rules in SNNs.

Despite increasing interest in pursuing high-performance SNNs with surrogate gradient training, there is limited understanding of how surrogate gradient training affects the representation of SNNs. Investigating this fundamental question is critical since surrogate gradient-based BPTT algorithm mimics the way how ANN learns and is less biological-plausible compared to other learning rules like STDP. Therefore, it would be intriguing to study whether surrogate gradient-based SNNs learn different representations than ANNs. Understanding the representation learned in SNN can also promote further research developments, e.g., designing spiking-friendly architectures (Kim et al., 2022a; Na et al., 2022) and exploring other ways to optimize SNNs (Bellec et al., 2020; Zhang & Li, 2020; Kim & Panda, 2021a).

More concretely, we ask, do SNNs optimized by surrogate gradient BPTT learn distinct representations from ANNs? How do the width and depth of the neural network affect the representation learned in SNNs and ANNs? Does the extra temporal dimension in SNNs yield unique intermediate features? On neuromorphic datasets, how does the SNN process event-based data? In this paper, we aim to answer these core questions through a detailed analysis of ResNets (He et al., 2016a) and VGG-series (Simonyan & Zisserman, 2015) models using a representation similarity analysis tool. Specifically, we utilize the popular Centered Kernel Alignment (CKA) (Kornblith et al., 2019) to measure the similarity between SNNs and ANNs. Fig. 1 demonstrates the overall workflow of our representation similarity analysis framework. Our analysis spans both the spatial and temporal dimensions of SNNs, as well as the impact of network architecture and input data.

Our contributions and findings include:

- We analyze the representation similarity between SNNs and ANNs using the centered kernel alignment to determine whether SNNs produce different feature representations from ANNs. We examine various aspects of representation similarity between SNNs and ANNs, including spatial and temporal dimensions, input data type, and network architecture.
- Surprisingly, our findings show that SNNs trained with surrogate gradient have a rather similar represen-tation to ANNs. We also find that residual connections greatly affect the representations in SNNs.

- Meanwhile, we find that the time dimension in SNNs does not provide much unique representation. We also find that shallow layers are insensitive to the time dimension, where the representation in each time step converges together.

## 2 Related Work

**Spiking Neural Networks (SNNs).** SNNs have gained increasing attention for building low-power intelligence. Generally, the SNN algorithms to obtain high performance can be divided into two categories: (1) ANN-SNN conversion (Rueckauer et al., 2016; 2017; Han et al., 2020; Sengupta et al., 2019; Han & Roy, 2020) and (2) direct training SNN from scratch (Wu et al., 2018; 2019). Conversion-based methods utilize the knowledge from ANN and convert the ReLU activation to a spike activation mechanism. This type of method can produce an SNN in a short time. For example, in Rueckauer et al. (2017), one can find the percentile number and set it as the threshold for spiking neurons. Authors in Deng & Gu (2021) and Li et al. (2021a) decompose the conversion error to each layer and then propose to reduce the error by calibrating the parameters. However, achieving near-lossless conversion requires a considerable amount of time steps to accumulate the spikes. Direct training from scratch allows SNNs to operate in extremely low time steps, even less than 5 (Zheng et al., 2020). To enable gradient-based learning, direct training leverages surrogate gradient to compute the derivative of the discrete spiking function. This also benefits the choice of hyper-parameters in spiking neurons. Recent works (Fang et al., 2021; Rathi & Roy, 2020; Kim & Panda, 2021b; Deng et al., 2022) co-optimize parameters, firing threshold, and leaky factor together via gradient descent. Our analysis is mostly based on directly trained SNNs, as converted SNNs only contain ANN features and may be misleading for representation comparison.

**Representation Similarity Analysis (RSA).** RSA (Kriegeskorte et al., 2008) was not originally designed for analyzing neural networks specifically. Rather, it is used for representation comparison between any two computational models. Prior works such as Khaligh-Razavi & Kriegeskorte (2014); Yamins et al. (2014) have used RSA to find the correlation between visual cortex features and convolutional neural network features. Authors of Seminar (2016); Raghu et al. (2017); Morcos et al. (2018); Wang et al. (2018) have studied RSA between different neural networks. However, recent work (Kornblith et al., 2019) argues that none of the above methods for studying RSA can yield high similarity even between two different initializations of the same architecture. They further propose CKA, which has become a powerful evaluation tool for RSA and has been successfully applied in several studies. For example, Nguyen et al. (2020) analyzes the representation pattern in extremely deep and wide neural networks, and Raghu et al. (2021) studies the representation difference between convolutional neural networks and vision transformers with CKA. In this work, we leverage this tool to compare ANNs and SNNs.

## 3 Preliminary

### 3.1 ANN and SNN Neurons

In this paper, vectors/matrices are denoted with bold italic/capital letters (*e.g.* $\boldsymbol{x}$ and $\mathbf{W}$ denote the input vector and weight matrix, respectively). Constants are denoted by small upright letters. For non-linear activation function in artificial neurons, we use the rectified linear unit (ReLU) (Krizhevsky et al., 2012), given by $\boldsymbol{y} = \max(0, \mathbf{W}\boldsymbol{x})$. As for the non-linear activation function in spiking neurons, we adopt the well-known Leaky Integrate-and-Fire (LIF) model. Formally, given a membrane potential $\boldsymbol{u}^{(t)}$ at time step $t$ and a pre-synaptic input $\boldsymbol{i}^{(t+1)} = \mathbf{W}\boldsymbol{x}^{(t+1)}$, the LIF neuron will update as

$$\boldsymbol{u}^{(t+1),\mathrm{pre}} = \tau\boldsymbol{u}^{(t)} + \boldsymbol{i}^{(t+1)}, \quad \boldsymbol{y}^{(t+1)} = \begin{cases} 1 & \text{if } \boldsymbol{u}^{(t+1),\mathrm{pre}} > v_{th} \\ 0 & \text{otherwise} \end{cases}, \quad \boldsymbol{u}^{(t+1)} = \boldsymbol{u}^{(t+1),\mathrm{pre}} \cdot (1 - \boldsymbol{y}^{(t+1)}). \quad (1)$$

Here, $\boldsymbol{u}^{(t+1),\mathrm{pre}}$ is the pre-synaptic membrane potential, $\tau$ is a constant leak factor within $(0,1)$. Let $v_{th}$ be the firing threshold, the LIF neuron will fire a spike ($\boldsymbol{y}^{(t+1)} = 1$) when the membrane potential exceeds the threshold; otherwise, it will stay inactive ($\boldsymbol{y}^{(t+1)} = 0$). After firing, the spike output $\boldsymbol{y}^{(t+1)}$ will propagate to

the next layer and become the input $\boldsymbol{x}^{(t+1)}$ of the next layer. Note that here the layer index is omitted for simplicity. The membrane potential will be reset to 0 if a spike fires (refer to Eq. (1) the third sub-equation).

## 3.2 Optimize SNN with Surrogate Gradient

To enable gradient descent for SNN, we adopt the BPTT algorithm (Werbos, 1990). Formally, denote the loss function value as $L$, the gradient of the loss value with respect to weights can be formulated by

$$\frac{\partial L}{\partial \mathbf{W}} = \sum_{t=1}^{T} \frac{\partial L}{\partial \boldsymbol{y}^{(t)}} \frac{\partial \boldsymbol{y}^{(t)}}{\partial \boldsymbol{u}^{(t),\text{pre}}} \mathbf{K}^{(t)}, \quad \text{where } \mathbf{K}^{(t)} = \left( \frac{\partial \boldsymbol{u}^{(t),\text{pre}}}{\partial \boldsymbol{i}^{(t)}} \frac{\partial \boldsymbol{i}^{(t)}}{\partial \mathbf{W}} + \frac{\partial \boldsymbol{u}^{(t),\text{pre}}}{\partial \boldsymbol{u}^{(t-1)}} \frac{\partial \boldsymbol{u}^{(t-1)}}{\partial \boldsymbol{u}^{(t-1),\text{pre}}} \mathbf{K}^{(t-1)} \right). \quad (2)$$

Here, the gradient is computed based on the output spikes from all time steps. In each time step, we denote $\mathbf{K}$ as the gradient of pre-synaptic membrane potential with respect to weights $\frac{\partial \boldsymbol{u}^{\text{pre}}}{\partial \mathbf{W}}$, which consists of the gradient of pre-synaptic input and the gradient of membrane potential in the last time steps.

As a matter of fact, all terms in Eq. (2) can be easily differentiated except $\frac{\partial \boldsymbol{y}^{(t)}}{\partial \boldsymbol{u}^{(t),\text{pre}}}$ which returns a zero-but-all gradient (Dirac delta function). Therefore, the gradient descent ends up either freezing the weights or updating weights to infinity. To address this problem, the surrogate gradient is proposed (Bender et al., 2018; Wu et al., 2018; Bellec et al., 2018; Neftci et al., 2019; Li et al., 2021b) to replace the Dirac delta function with another function:

$$\frac{\partial \boldsymbol{y}^{(t)}}{\partial \boldsymbol{u}^{(t),\text{pre}}} = \frac{1}{\alpha} \mathbb{1}_{|\boldsymbol{u}^{(t),\text{pre}} - v_{th}| < \alpha}, \quad (3)$$

where $\alpha$ is a hyper-parameter for controlling the sharpness and $\alpha = 1$ the surrogate gradient becomes the Straight-Through Estimator (Bengio et al., 2013).

Compared to other methods such as ANN-SNN conversion (Deng & Gu, 2021) or spike-timing-dependent plasticity (Caporale et al., 2008), BPTT using surrogate gradient learning yields the best performance in image recognition tasks. However, from a biological perspective, BPTT is implausible: for each weight update, BPTT requires the use of the transpose of the weights to transmit errors backward in time and assign credit for how past activity affected present performance. Running the network with transposed weights requires the network to either have two-way synapses or use a symmetric copy of the feedforward weights to backpropagate the error (Marschall et al., 2020). Therefore, the question remains whether the representation in SNNs learned with surrogate gradient-based BPTT actually differs from the representation in ANNs.

## 3.3 Centered Kernel Alignment

Let $\mathbf{X}_s \in \mathbb{R}^{m \times Tp_1}$ and $\mathbf{X}_a \in \mathbb{R}^{m \times p_2}$ contain the representation in an arbitrary layer of SNN with $p_1$ hidden neurons across $T$ time steps and the representation in an arbitrary layer of ANN with $p_2$ hidden neurons, respectively. Here $m$ is the batch size and we concatenate features from all time steps in the SNN altogether. We intend to use a similarity index $s(\mathbf{X}_s, \mathbf{X}_a)$ to describe how similar they are. We use the Centered Kernel Alignment (CKA) (Kornblith et al., 2019) to measure this:

$$\text{CKA}(\mathbf{K}, \mathbf{L}) = \frac{\text{HSIC}(\mathbf{K}, \mathbf{L})}{\sqrt{\text{HSIC}(\mathbf{K}, \mathbf{K})\text{HSIC}(\mathbf{L}, \mathbf{L})}}, \quad \text{HSIC}(\mathbf{K}, \mathbf{L}) = \frac{1}{(m-1)^2} tr(\mathbf{K}\mathbf{H}\mathbf{L}\mathbf{H}). \quad (4)$$

Here, $\mathbf{K} = \mathbf{X}_s\mathbf{X}_s^{\top}, \mathbf{L} = \mathbf{X}_a\mathbf{X}_a^{\top}$ are the Gram matrices as shown in Fig. 1. Each Gram matrix has the shape of $m \times m$, reflecting the similarities between a pair of examples. For example, $\mathbf{K}_{i,j}$ indicates the similarity between the $i^{th}$ and $j^{th}$ example in the SNN feature $\mathbf{X}_s$. Further measuring the similarity between $\mathbf{K}$ and $\mathbf{L}$, one can measure whether SNN has a similar inter-example similarity matrix with ANN. Let $\mathbf{H} = \mathbf{I} - \frac{1}{m}\mathbf{1}\mathbf{1}^{\top}$ be the centering matrix, the Hilbert-Schmidt Independence Criterion (HSIC) proposed by Gretton et al. (2005) can conduct a test statistic for determining whether two sets of variables are independent. HSIC = 0 implies independence. The CKA further normalizes HSIC to produce a similarity index between 0 and 1 (the higher the CKA, the more similar the input pair) which is invariant to isotropic scaling. In our implementation, we use the unbiased estimator of HSIC (Song et al., 2012; Nguyen et al., 2020) to calculate it across mini-batches.

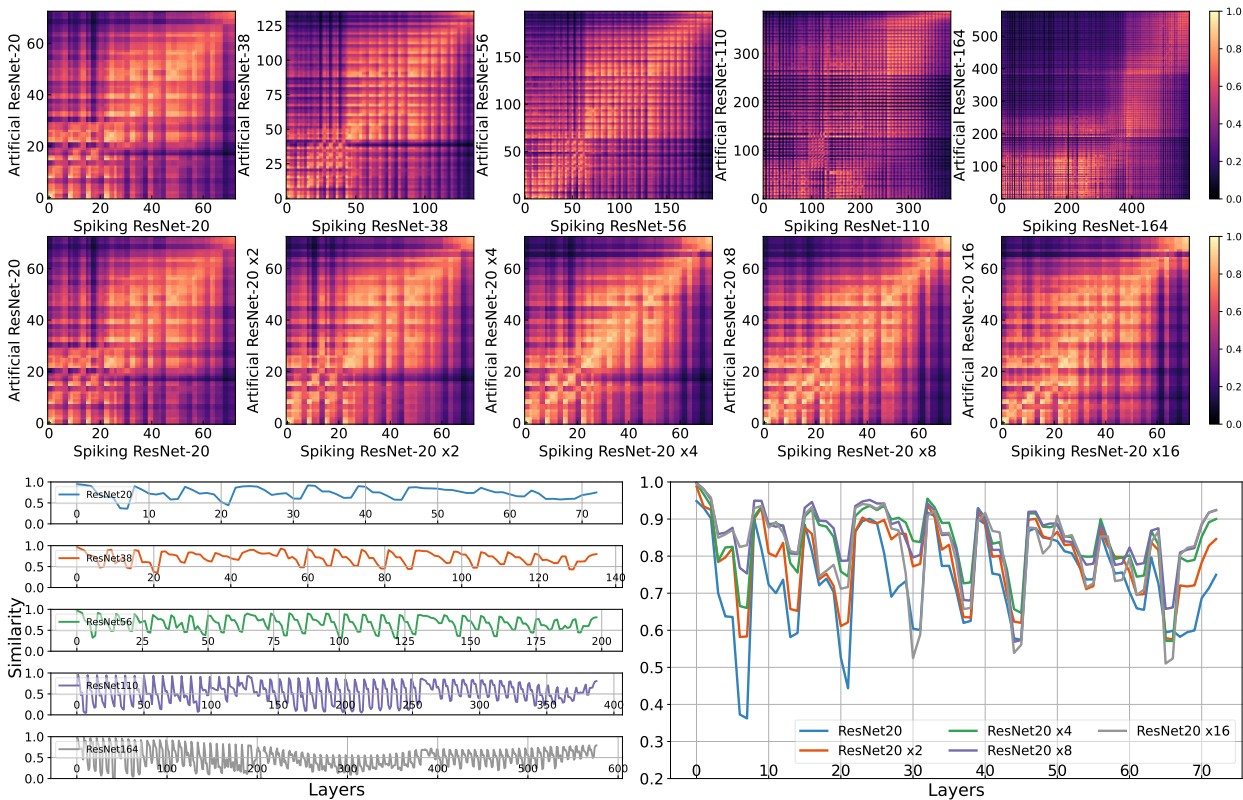

Figure 2: **CKA heatmap between SNNs and ANNs with different depth and width on the CIFAR-10 dataset.** **Top**: the CKA cross-layer heatmap across different depths from 20 layers to 164 layers. **Middle**: the CKA cross-layer heatmap across different widths from the original channel number to 16 times. **Bottom**: visualizing only the corresponding layer, which is the diagonal of the CKA heatmap. We find generally SNNs and ANNs have relatively high similarity, and deeper/wider networks have positive/negative effects on the representation similarity.

## 4 Do SNNs Learn Different Representation from ANNs?

In this section, we comprehensively compare the representation learned in SNNs and ANNs. Our primary study case is ResNet with identity mapping block (He et al., 2016b) on the CIFAR10 dataset, which is the standard architecture and dataset in modern deep learning for image recognition. [1] There are two differences between our SNNs and ANNs. First, ANNs adopt the Batch Normalization layer (Ioffe & Szegedy, 2015), and SNNs use the time-dependent Batch Normalization layer (Zheng et al., 2020), which normalizes the feature across all time steps (*i.e.* $\mathbf{X}s$). Second, the ANNs use ReLU activation, and SNNs leverage the LIF spiking neurons. For default SNN training, we use direct encoding, $\tau = 0.5$ for the leaky factor, $vth = 1.0$ for the firing threshold, $T = 4$ for the number of time steps, and $\alpha = 1.0$ for surrogate gradient, which are tuned for the best training performance on SNNs. Detailed training setup and codes can be found in the supplementary material.

### 4.1 Scaling up Width or Depth

We begin our study by studying how the spatial dimension of a model architecture affects internal representation structure in ANNs and SNNs. We first investigate a simple model: ResNet-20, and then we either increase its number of layers or increase its channel number to observe the effect of depth and width, respectively. In the most extreme cases, we scale the depth to 164 and the width to 16× (see detailed network configuration in Table D.1). For each network, we compute CKA between all possible pairs of layers, includ-

---

[1] We also provide RSA on VGG-series networks in Sec. A.1 and RSA on CIFAR100 dataset in Sec. A.2.

Table 1: The top-1 accuracy of SNNs and ANNs on CIFAR-10 dataset, as well as the gap $\Delta$ between ANN and SNN.

| Network | Depth | | | | | Width | | | | |
|---|---|---|---|---|---|---|---|---|---|---|
| | 20 | 38 | 56 | 110 | 164 | $\times 1$ | $\times 2$ | $\times 4$ | $\times 8$ | $\times 16$ |
| ANN | 91.06 | 92.34 | 92.98 | 92.37 | 93.00 | 91.06 | 93.36 | 94.52 | 94.78 | 94.78 |
| SNN | 89.67 | 91.14 | 91.94 | 91.83 | 92.05 | 89.67 | 92.00 | 93.96 | 94.48 | 94.73 |
| $\Delta_{\text{A-S}}$ | 1.39 | 1.20 | 1.04 | 0.53 | 0.95 | 1.39 | 1.36 | 0.56 | 0.30 | 0.05 |

ing convolutional layers, normalization layers, ReLU/LIF layers, and residual block output layers. Therefore, the total number of layers is much greater than the stated depth of the ResNet, as the latter only accounts for the convolutional layers in the network. Then, we visualize the result as a heatmap, with the $x$ and $y$ axes representing the layers of the network, going from the input layer to the output layer. Following Nguyen et al. (2020), our CKA heatmap is computed on 4096 images from the test dataset.

As shown in Fig. 2, the CKA heatmap emerges as a checkerboard-like grid structure, especially for the deeper neural network. In ResNet-20, we observe a bright block in the middle and deep layers, indicating that ANNs and SNNs learn overlapped representation. As the network goes deep, we find the CKA heatmap becomes darker, meaning that representations in ANNs and those in SNNs are diverging. Notably, we find a large portion of layers in artificial ResNet-164 exhibit significantly different representations than spiking ResNet-164 (<0.2 CKA value) which demonstrates that deeper layers tend to learn disparate features.

In Fig. 2 middle part, we progressively enlarge the channel number of ResNet-20. In contrast to depth, the heatmap of wider neural networks becomes brighter, which indicates the representations in SNNs and ANNs are converging. Interestingly, although the majority of layers are learning more similar representations between ANN and SNN in wide networks, the last several layers still learn different representations.

We further select only the diagonal elements in the heatmap and plot them in Fig. 2 bottom part. Because SNNs and ANNs have the same network topology, this visualization is more specific and may accurately reveal the similarity between SNNs and ANNs at each corresponding layer. First, we can find that in Fig. 2 the CKA curve of ResNet-20 shows relatively high values. Most layers go above 0.5 and some of them can even reach nearly 1.0. Interestingly, we observe that deeper networks tend to derive a curve with a jagged shape. This means some layers in SNN indeed learn different representations when compared to ANN, however, *the difference is intermittently mitigated.* In later sections, we will show that the mitigation of dissimilarity is performed by residual connection. As for width, we generally notice that CKA curves mostly become higher for wider networks, especially when comparing ResNet-20 and ResNet-20 $\times 8$, where most layers have above 0.8 CKA value.

**Evaluating the Task Performance.** We lay out the accuracy of ANNs and SNNs in Table 1. Additionally, we also calculate their accuracy difference across different width and depth configurations. We can find that the accuracy gap decreases if we scale up the width, ranging from 1.39% to 0.05%. However, as the depth increases, the accuracy gap does not consistently reduce as it does in wider networks. This explains that wider neural networks which bring more similar representations encourage less accuracy gap. However, SNNs do not achieve the same accuracy level as ANNs with different representations. This finding means that current state-of-the-art SNNs relying on wider neural networks (e.g., ResNet-19 (Zheng et al., 2020) which is similar to our ResNet-20 8$\times$) do not truly develop unique feature representations than ANNs.

## 4.2 The Effect of Residual Connections

In Fig. 2, the CKA curves appear with a periodic jagged shape. To investigate what causes this similarity oscillation, we investigate each layer in a residual block. In Fig. 3 left, we plot the CKA curve of the ResNet-110 and additionally sample two residual blocks, the 10-*th* and the 34-*th* block, whose architecture details are depicted in Fig. 3 right. Surprisingly, we find that every time when the residual connection meets the main branch, the CKA similarity restores nearly to 1. Moreover, every time when the activation passes a convolutional layer or an LIF layer, the similarity decreases. The BN layers, in contrast, do not affect the

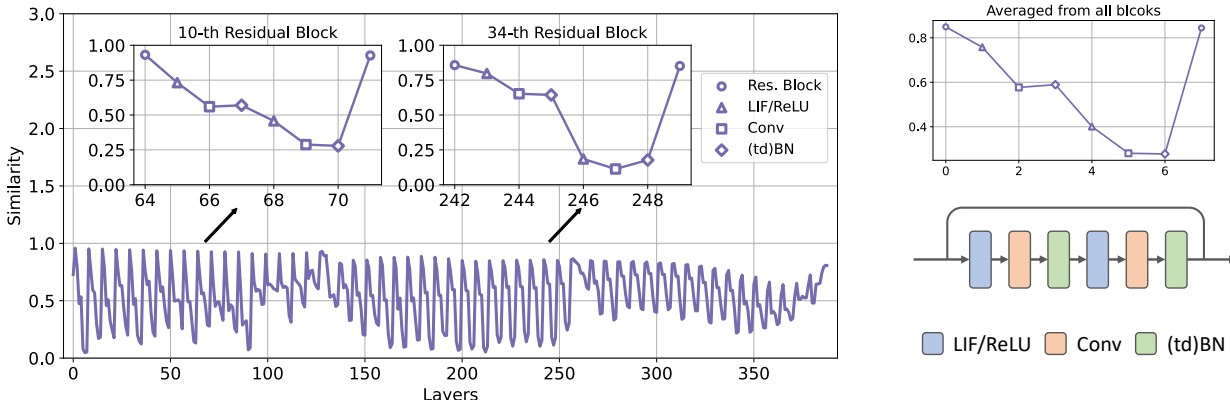

Figure 3: **Emergence of periodic jagged CKA curve. Left**: CKA curve of ResNet-110. We subplot the 10-*th* and the 34-*th* residual blocks in ResNet-110, which forms a periodic jagged curve. **Top right**: The averaged CKA value in all blocks, **Bottom right**: The architecture specification of the residual block we used.

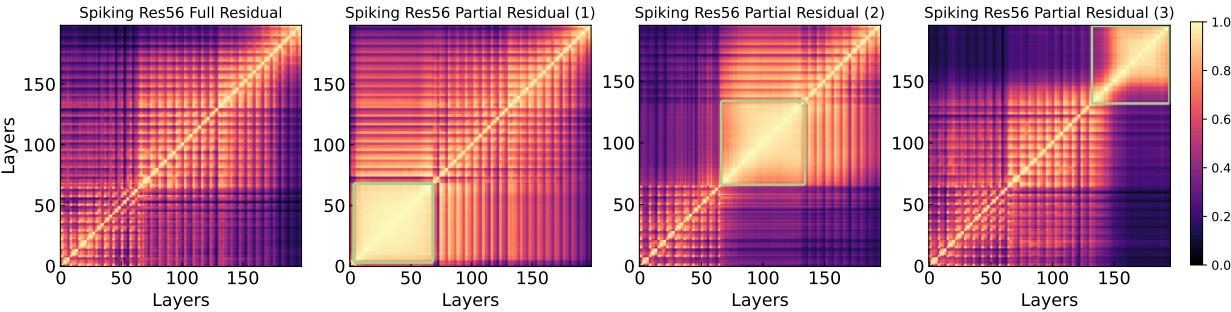

Figure 4: **The effect of residual connections in SNNs.** We remove residual connections in one of three stages in the ResNet-56 and show the CKA heatmaps. The non-residual stage is annotated with green square □.

similarity since it is a linear transformation. These results substantiate that *the convolutional layers and LIF layers in SNNs are able to learn different representations than ANNs. However, the representation in the residual branch still dominates the representation in post-residual layers and leads to the junction of ANN's and SNN's representation.*

To further explore why residual connections can restore the representations in SNNs to ANN-like, we conduct an ablation study. We select one of the three stages in the spiking ResNet-56 where the residual connections are disabled selectively. In Fig. 4, we visualize the CKA heatmaps of SNN itself, which means both $x$ and $y$ axes are the same layers in SNN. The first heatmap demonstrates the full residual network, while the remaining three heatmaps show the partial residual networks, with the 1st, 2nd, and 3rd stages disabled, respectively. Our observations can be summarized as follows: (1) In terms of inter-stage similarity, residual connections can preserve the input information from the previous stage. In the 1st and 2nd heatmaps in Fig. 4, we find residual blocks can have high similarity with their former stage. The non-residual block, however, does not have this property. In the 3rd and 4th heatmaps, we can see that blocks without residual connections exhibit significantly different representations when compared to their former stage. Therefore, we can find that residual connections preserve the representation in early layers. As such, if ANN and SNN learn similar representations in the first layer, the similarity can propagate to very deep layers due to residual connections. (2) In terms of intra-stage similarity, the non-residual stage's heatmap appears with a uniform representation across all layers, meaning that layers in this stage are similar. In contrast, residual stages share a grid structure.

Next, we verify the accuracy of SNNs and ANNs when both are equipped with residual connections or not, under different network depths. As shown in Table 2, both the SNNs and ANNs can successfully train very

Table 2: **The impact of residual connections on accuracy.**

| Model | Type | Depth | | | |
|---|---|---|---|---|---|
| | | 20 | 38 | 56 | 74 |
| ANN | w/. residual connection | 91.06 | 92.34 | 92.98 | 92.85 |
| | w/o residual connection | 91.32 | 91.17 | 89.62 | 21.07 |
| SNN | w/. residual connection | 89.63 | 91.14 | 91.94 | 91.83 |
| | w/o residual connection | 86.50 | 82.64 | 33.61 | 10.00 |

deep networks if the residual connections are enabled. In this case, though SNNs do not surpass the accuracy of ANNs, the gap is relatively small, with 1∼2% accuracy degradation. However, if the residual connections are removed from SNNs, the gap between the accuracies of ANNs and SNNs significantly enlarges, ranging from 5∼56%. Therefore, we can conclude that *the residual connections help the gradient descent optimization in SNNs and regularize the representations in SNNs to be similar to those in ANNs so that SNNs can have similar task performances with ANNs.*

### 4.3 Scaling up Time Steps

The results of the previous sections help characterize the effects of spatial structure on internal representation differences between SNNs and ANNs. Next, we ask whether the time dimension helps SNN learn some unique information. To verify this, we train several spiking ResNet-20 with 4/8/16/32/64/128 time steps and calculate the ANN-SNN CKA similarity. In Fig. 5, we envision the CKA heatmaps and curves respectively between artificial ResNet-20 and spiking ResNet-20 with various time steps. Notably, we cannot find significant differences among these heatmaps. Looking at the CKA curves, we also discover that many layers are overlapped, especially when we focus on the residual block output (the local maximums). We find similarities between different time steps reaching the same point, meaning that the time step variable does not provide much unique representation in SNNs.

To further analyze the representation along the time dimension in SNNs, we compare the CKA among various time steps. Concretely, for any layer inside an SNN, we reshape the feature $\mathbf{X}_s$ to $[\mathbf{X}^{(1)}, \mathbf{X}^{(2)}, \dots, \mathbf{X}^{(T)}]$ where $\mathbf{X}^{(i)}$ is the *i-th* time step's output. By computing the CKA similarity between arbitrary two time steps, *i.e.,* $\text{CKA}(\mathbf{X}^{(i)}, \mathbf{X}^{(j)})$, we are able to construct a CKA heatmap with $x, y$ axes being the time dimension, which demonstrates whether the features are similar across different time steps. Fig. 6 illustrates such CKA heatmaps of outputs from all residual blocks in the spiking ResNet-20, with time steps varying from 4 to 32. In general, deeper residual block output exhibits darker CKA heatmaps and the shallower layers tend to become yellowish. In particular, all the residual blocks from the first stage have an all-yellow CKA heatmap,

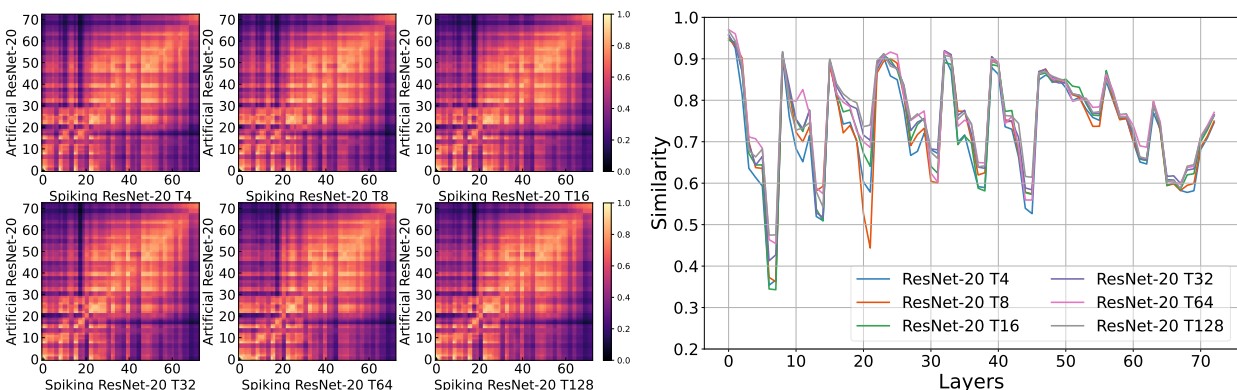

Figure 5: **The effect of time steps in SNNs. Left**: CKA heatmaps between ANNs and SNNs with the different number of time steps. **Right**: The CKA curve of corresponding layers (diagonal values as in left).

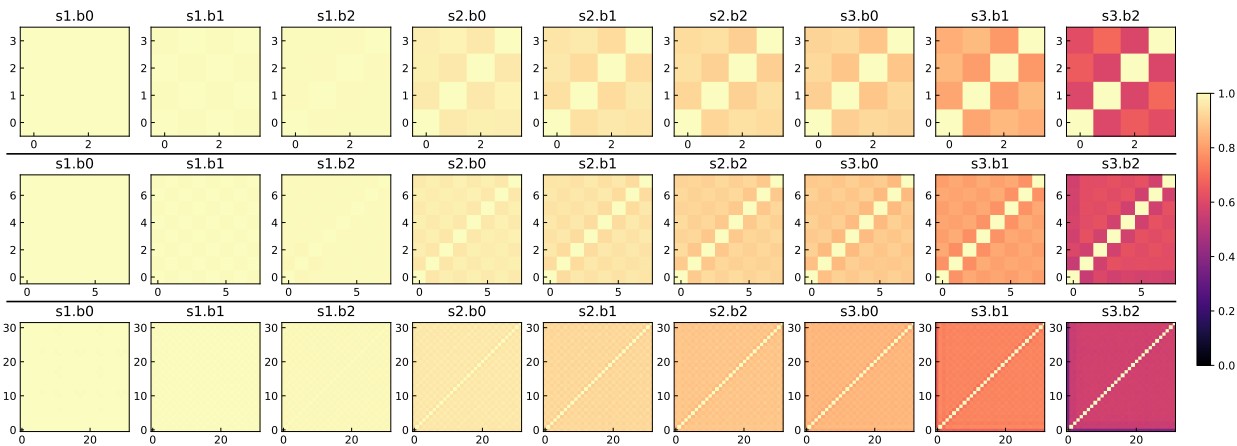

Figure 6: **The similarity across times in SNN.** Each heatmap shows the CKA among different time steps in the output of the residual block. "s" means stage, and "b" means block. The top/middle/bottom rows are spiking ResNet-20 with 4/8/32 time steps, respectively.

Table 3: **The sensitivity of time steps in SNNs.**

| Model | Type | # Time Steps | | | |
|-------|------|------|------|------|------|
| | | 4 | 8 | 16 | 32 |
| | Full time steps | 89.67 | 90.44 | 90.98 | 90.99 |
| SNN | Reduce the time steps in the first stage | **88.81** | **89.70** | **89.91** | **90.47** |
| | Reduce the time steps in the last stage | 87.41 | 87.78 | 87.38 | 87.73 |

indicating extremely high similarity in these blocks. The second stage starts to produce differences across time steps, but they still share >0.8 similarities between any pair of time steps. The last stage, especially the last block, demonstrates around 0.5 similarities between different time steps. To summarize, the impact of time in SNN is gradually increased as the feature propagates through the network. In Appendix A.4, we provide the heatmap of convolutional/LIF layers and find a similar trend.

We further conduct an empirical verification to verify the findings in Fig. 6. More specifically, we define a sensitivity metric and measure it by reducing the number of time steps to 1 in certain layers of an SNN and recording the corresponding accuracy degradation. In Fig. 6 we find the first stage (s1) has the same representation in time dimension while the last stage (s3) exhibits a more diverse representation. Therefore, we choose to reduce the number of time steps to 1 either in s1 or in s3. To achieve this "mixed-time-step SNN", we repeat/average the activation in time dimension after s1/before s3 to match the dimension. Table 3 summarizes the sensitivity results. We observe that the first stage can have much lower accuracy degradation (<1%), while the last stage drop 2∼4% accuracy. Moreover, if the last stage only uses 1 time step, then increasing the time steps for the other two stages does not benefit the accuracy at all. This indicates that LIF neurons are more effective in deep layers than in shallow layers.

### 4.4 Representation under Event Data

In this section, we evaluate the CKA similarity on the event-based dataset. We choose CIFAR10-DVS (Li et al., 2017), N-Caltech 101 (Orchard et al., 2015) and train spiking/artificial ResNet-20. Since the ANN cannot process 4-dimensional spatial-temporal event data easily, we integrate all events into one frame for ANN training and 10 frames for SNN training. Fig. 7 provides the CKA heatmaps/curves on the event dataset, showing a different similarity distribution than the previous CIFAR-10 dataset. The heatmaps have a different pattern and the curves also do not appear with a periodic jagged shape. In addition, the similarity distribution differs at the dataset level, i.e., the CIFAR10-DVS and N-Caltech 101 share different

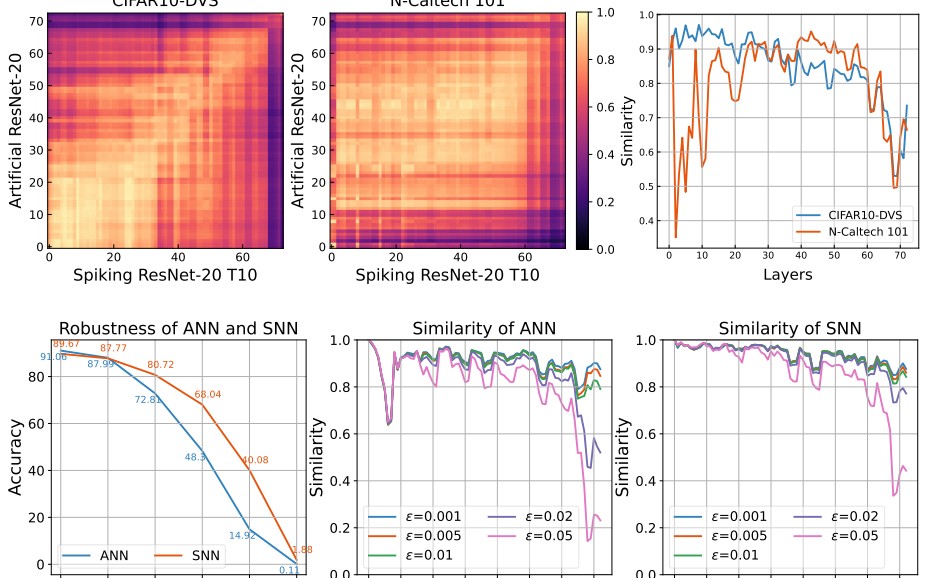

Figure 7: **CKA Similarity on Event Dataset.** We train spiking and artificial ResNet-20 on CIFAR10-DVS and N-Caltech 101, respectively. **Left**: the CKA heatmaps. **Right**: The CKA curves of corresponding layers between ANN and SNN.

Figure 8: **The robustness against adversarial attack.** **Left**: The accuracy of SNN and ANN after attack under different $\epsilon$. **Right**: The CKA curve between clean images and adversarial images of ANN and SNN, respectively.

CKA curves and heatmaps. On N-Caltech 101, the SNN learns different feature representations compared with ANN in shallow and deep layers, but similar representation in intermediate layers. For CIFAR10-DVS, the similarity continues to decrease from 0.9 to 0.5 as the layers deepen. In summary, with the event-based dataset, SNNs and ANNs share a different CKA pattern in comparison to the natural image dataset, which implies that SNNs may have further optimization space in this type of dataset. We put more CKA results on various models for the CIFAR10-DVS dataset in Appendix A.3.

## 4.5 Representation under Adversarial Attack Data

We next study the adversarial robustness of SNN and ANN using CKA. Inspired by quantized ANNs that are robust to adversarial attack Lin et al. (2019), the SNNs could also inherit this property since their activations are also discrete. Previous works have explored understanding the inherent robustness of SNNs (Sharmin et al., 2020; Kundu et al., 2021; Liang et al., 2021; Kim & Panda, 2021c). However, they either evaluate on converted SNN or using rate-encoded images. Here, we test Projected Gradient Descent (PGD) attack (Madry et al., 2017) on the directly trained SNN and ANN using direct encoding. Formally, we generate the adversarial images by restricting the $L$-infinity norm of the perturbation, given by

$$\boldsymbol{x}_{adv}^{k+1} = \Pi_{P_\epsilon(\boldsymbol{x})}(\boldsymbol{x}_{adv}^k + \alpha\mathrm{sign}(\nabla_{\boldsymbol{x}_{adv}^k} L(\boldsymbol{x}_{adv}^k, \boldsymbol{w}, \boldsymbol{y}))), \tag{5}$$

where $\boldsymbol{x}_{adv}^k$ is the generated adversarial sample at the $k$-th iteration. $\Pi_{P_\epsilon(\boldsymbol{x})}(\cdot)$ projects the generated sample onto the projection space, the $\epsilon - L_\infty$ neighborhood of the clean sample. $\alpha$ is the attack optimization step size. With higher $\epsilon$, the adversarial image is allowed to be perturbed in a larger space, thus degrading task performance.

We evaluate the performance of spiking and artificial ResNet-20 on the CIFAR-10 dataset with $\epsilon$ values ranging from $0.001, 0.005, 0.01, 0.02, 0.05$ to generate adversarial images. We then compute the CKA value between the features of the clean images and the adversarially corrupted images. The results are summarized in Fig. 8 (left). We find that although the clean accuracy of ANN is higher than that of SNN, SNN has higher robustness against adversarial attacks. For example, the PGD attack with 0.01 $L$-infinity norm perturbation reduces the accuracy of ANN by 43%, while only reducing the accuracy of SNN by 22%. We also investigate the CKA similarity between clean and adversarial images, as shown in the second and third subplots of Fig. 8. We observe that the higher the robustness against adversarial attacks, the higher the similarity between clean and corrupted images. This intuition is confirmed by the CKA curves, which show that SNN has a higher similarity than ANN. We also observe several interesting phenomena. For example, the ANN

suffers a large decrease in similarity in the first block, even with a small $\epsilon$ value of 0.001. Additionally, when we focus on the purple line ($\epsilon = 0.02$), we notice that ANN and SNN have similar perseverance in earlier layers, but ANN drops much more similarity than SNN in the last block. These results provide insight into model robustness and suggest that SNNs are more robust than ANN, especially in their shallow and deep layers.

## 5 Discussion and Conclusion

Given that SNNs are drawing increasing research attention due to their bio-plausibility and recent progress in task performance, it is necessary to verify if SNNs, especially those trained with the surrogate gradient algorithms, can or have the potential to truly develop desired features different from ANNs. In this work, we conduct a pilot study to examine the internal representation of SNNs and compare it with ANNs using the popular CKA metric. This metric measures how two models respond to two different examples. Our findings can be briefly summarized as follows:

1. Generally, the layer-wise similarity between SNNs and ANNs is high, suggesting SNNs trained with surrogate gradient learn similar representation with ANNs. Moreover, wider networks like ResNet-20 $8\times$ can even have $> 0.8$ similarities for almost all layers.
2. For extremely deep ANNs and SNNs, the CKA value would become lower, however, the residual connections play an important role in regularizing the representations. By conducting ablation studies, we demonstrate that the residual connections make SNNs learn similar representations with ANNs and help SNNs achieve high accuracy.
3. The time dimension does not provide much additional representation power in SNNs. We also demonstrate that the shallow layers learn completely static representation along the time dimension. Even reducing the number of time steps to 1 in shallow layers does not significantly affect the performance of SNNs.
4. On other types of datasets, SNNs may develop less similar representations with ANNs, *e.g.,* event-data.

Our results show that SNNs optimized by surrogate gradient algorithm do not learn distinct spatial-temporal representation compared to the spatial representation in ANNs. Current SNN learning relies on the residual connection and wider neural networks (for example, Zheng et al. (2020) use ResNet-19 which is similar to our ResNet-20 $8\times$) to obtain decent task performance. However, our study suggests that this task performance is highly credited to the similar representation with ANN. Furthermore, the time dimension brings limited effect to the SNN representation on static datasets like CIFAR10 and CIFAR100. In particular, the first stage of ResNets results in quite similar representation across time steps.

Nonetheless, our study is not a negation against SNNs. Our results are based on the surrogate-gradient BPTT optimization, which, as aforementioned, is inherently bio-implausible and resembles the optimization method for ANNs. Therefore, it may not be surprising to see SNNs and ANNs have similar representations under a similar optimization regime. Additionally, we find that input data is also important in developing the representations. Indeed, the direct encoding used in SNNs inputs the same static images for each time step, again leading to less gap between the representations of ANNs and SNNs.

Here, we provide several directions worth studying in the future: a) Bio-plausible learning rule for SNNs: surrogate gradient training tends to learn an ANN-like representation in SNN, thus it is necessary to develop an optimization method that suits SNN better. b) Spiking architecture design: a specialized SNN network architecture may avoid learning similar representation, *e.g.,* Kim et al. (2022a). c) Understanding the robustness of SNN: adversarial attack is inconsequential for human visual systems, which may be reflected in SNN as well. We believe the SNN robustness can be significantly improved. To conclude, our work tries to understand the representation of SNNs trained with surrogate gradient and reveals some counter-intuitive observations. We hope our work can inspire more research in pushing the limit of SNNs.

**Acknowledgements.** This work was supported in part by CoCoSys, a JUMP2.0 center sponsored by DARPA and SRC, Google Research Scholar Award, the National Science Foundation CAREER Award, TII (Abu Dhabi), the DARPA AI Exploration (AIE) program, and the DoE MMICC center SEA-CROGS (Award #DE-SC0023198).

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

# A   Additional CKA Results

## A.1   Results on VGG Networks

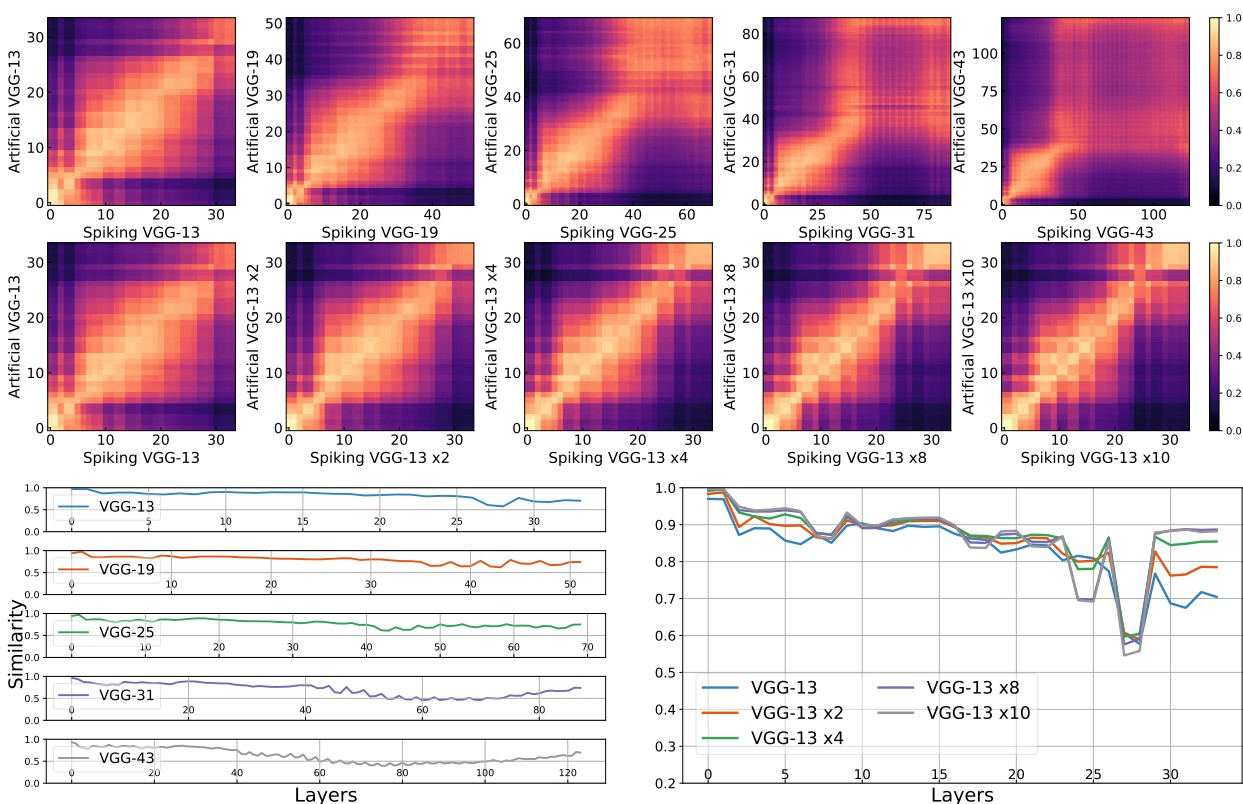

Figure A.1: **CKA heatmap between spiking VGGs and artificial VGGs with different depth and width on CIFAR-10 dataset. Top**: the CKA cross-layer heatmap across different depth from 13 layers to 43 layers. **Middle**: the CKA cross-layer heatmap across different width from original channel number to 8 times. **Bottom**: visualizing only the corresponding layer, which is the diagonal of the CKA heatmap.

### A.1.1   Scaling up Width or Depth

In this section, we study the representation similarity between ANNs and SNNs based on VGG-series networks Simonyan & Zisserman (2015). Since VGG-Networks do not employ residual connections, they may bring different representation heatmaps when compared to ResNets. We first evaluate if the spatial scaling observation in ResNets can also be found in VGG Networks. Starting from a VGG-13, we either increase its channel size to 10 times as before or its number of layers to 43, (detailed network configuration is provided in Table D.2). Since VGG networks do not contain residual connections, we could scale less depth in VGG networks than ResNets.

The results are illustrated in Fig. A.1. We can find that for deeper networks, the heatmaps tend to exhibit a hierarchical structure, which means the shallow layers and deeper layers have different representations. Increasing the number of layers in VGG networks to 19 or 25, the shallower and deeper layers only have 0.3 CKA similarity (purple). More seriously, when the network depth reaches 31 or 43, the similarity becomes 0.4 even across each other in deeper layers, indicating the representations are diverging. Another important discovery for deep VGG networks is the disappearance of periodical CKA curve, which is likely due to the lack of residual connections.

As for the wider networks, the observations are consistent with ResNet families. The wider networks have both brighter heatmaps and higher CKA curves. These results confirm the similar representation in extremely wide networks.

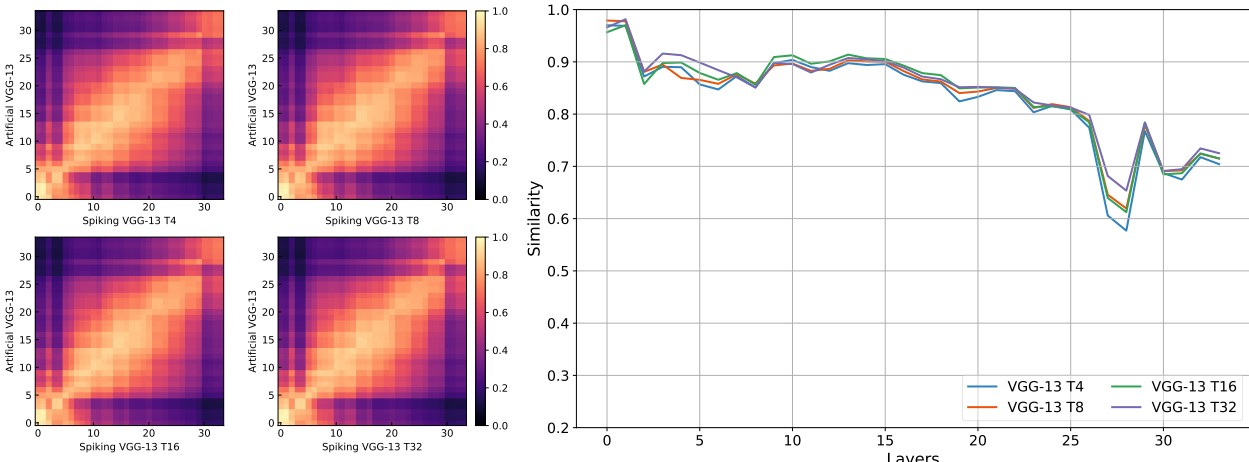

Figure A.2: **The effect of time steps in spiking VGG networks. Left**: CKA heatmaps between ANNs and SNNs with the different number of time steps (from 4 to 32). **Right**: The CKA curve of corresponding layers (diagonal values as in left).

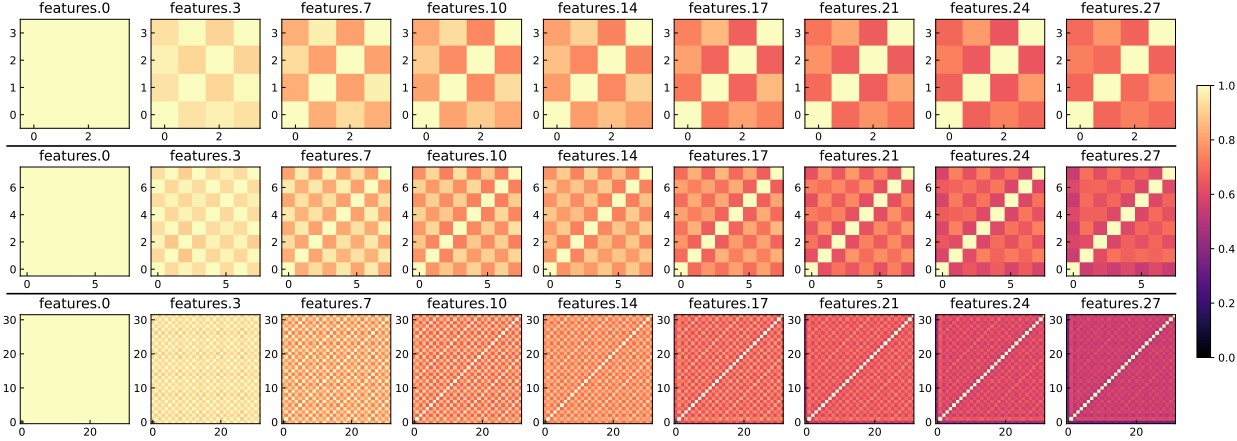

Figure A.3: **The similarity across times in Spiking VGG-13.** Each heatmap shows the CKA among different time steps in the output of convolutional layers. The top/middle/bottom rows stand for spiking ResNet-20 with 4/8/32 time steps.

### A.1.2   Scaling up Time Steps

We next study whether the time dimension in spiking VGG networks has a similar effect to that in spiking ResNets. As can be found in Fig. A.2, the spiking VGG-13s with 4/8/16/32 time steps do not show a significant difference in CKA heatmaps as well as CKA curves, which is the same with Fig. 5. Fig. A.3 shows the CKA across different time steps in the spiking VGGs. In the first layer, the convolution does not have dynamic representation through time. As the layer goes deeper, the dissimilarity across time steps continues to increase, similar to Fig. 6.

## A.2    Results on CIFAR-100

The results we reported in the main context are majorly based on the CIFAR-10 dataset. Here, we provide the visualizations on the CIFAR-100 dataset to further strengthen our findings in the main context.

We first report the spatial dimension results, i.e., scaling up the width and depth of the network. Starting from the ResNet-20, we either increase its width to 164 layers or increase its width to 8 times as before. The visualizations are shown below. We find extremely deep networks, e.g., ResNet-164, has a very dark heatmap compared to the ResNet-20 heatmap. The wider network shows somewhat irregular results. The extremely wide network — ResNet-20×8 — yet has even lowest similarity in the last several blocks. However, it has the highest similarity in the output layer.

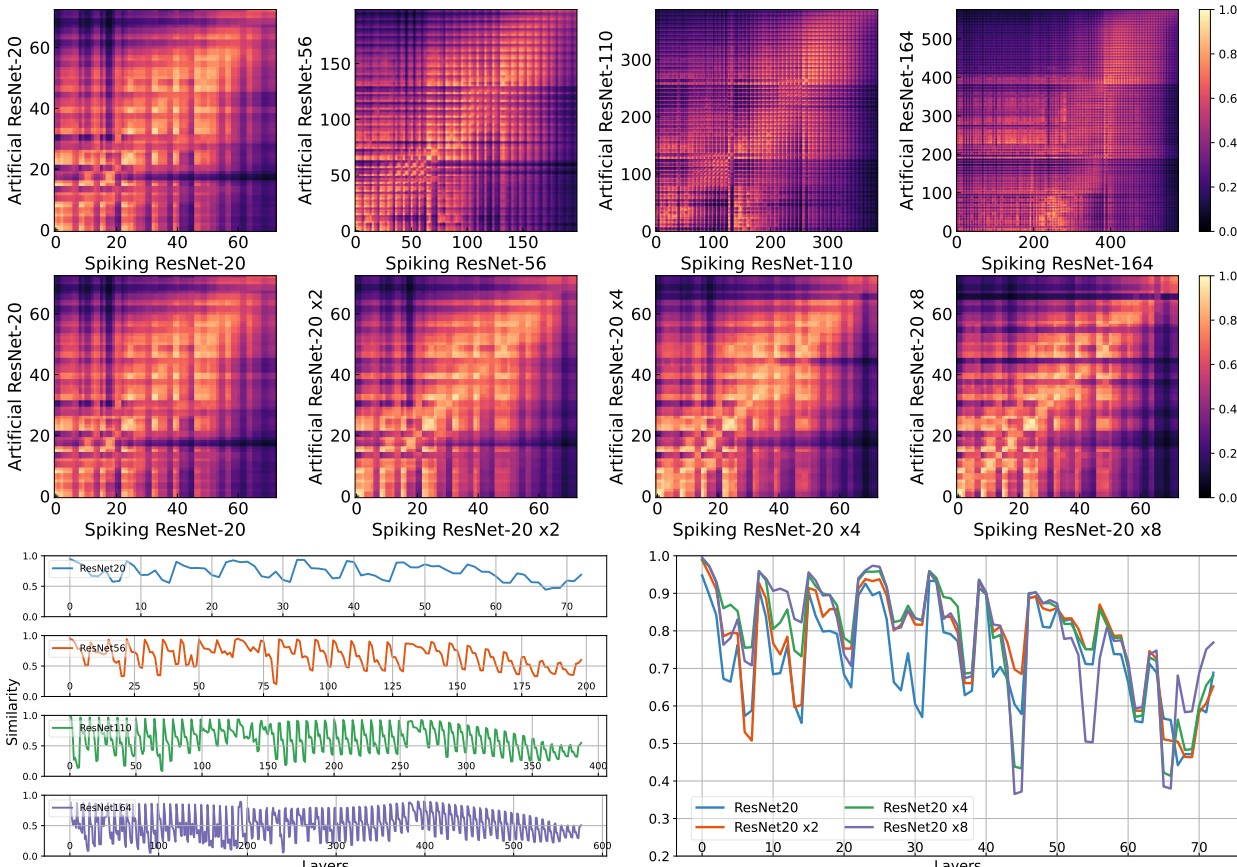

Figure A.4: **CKA heatmap between SNNs and ANNs with different depth and width on CIFAR-100.** **Top**: the CKA cross-layer heatmap across different depth from 20 layers to 164 layers. **Middle**: the CKA cross-layer heatmap across different width from original channel number to 16 times. **Bottom**: visualizing only the corresponding layer, which is the diagonal of the CKA heatmap.

Next, we visualize the details inside a residual block. In Fig. A.5, we sub-sample the 10-*th* and the 34-*th* residual block in a ResNet-110, which shows the same phenomenon. The LIF and convolutional layers cause a decrease in similarity, while the residual addition operation restores the similarity. We also provide the CKA heatmap of the partial residual network. As done in Fig. 4, we train 3 spiking ResNet-56 on the CIFAR-100 dataset with several blocks disabling the residual connections. Moreover, we train a linear probing layer — the fully-connected classifier on top of each block to see if it contributes to the overall performance of the whole network. The visualization is shown in Fig. A.6, where we find similar observations.

We also run experiments to test the time dimension of SNNs on CIFAR-100. In Fig. A.7, we train 4 spiking ResNet-20 with 4/8/16/32 time steps and compute their representations with artificial ResNet-20. Both

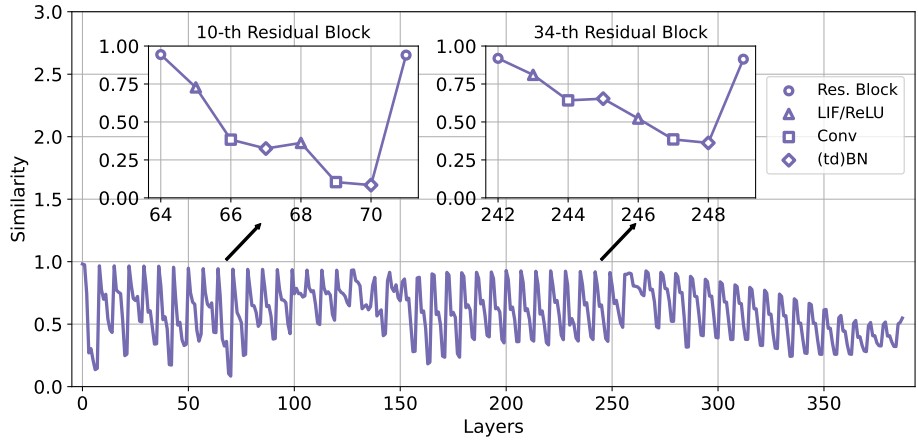

Figure A.5: **Emergence of periodic jagged CKA curve on CIFAR-100.** We subplot the 10-*th* and the 34-*th* residual blocks in ResNet-110, which forms a periodic jagged curve.

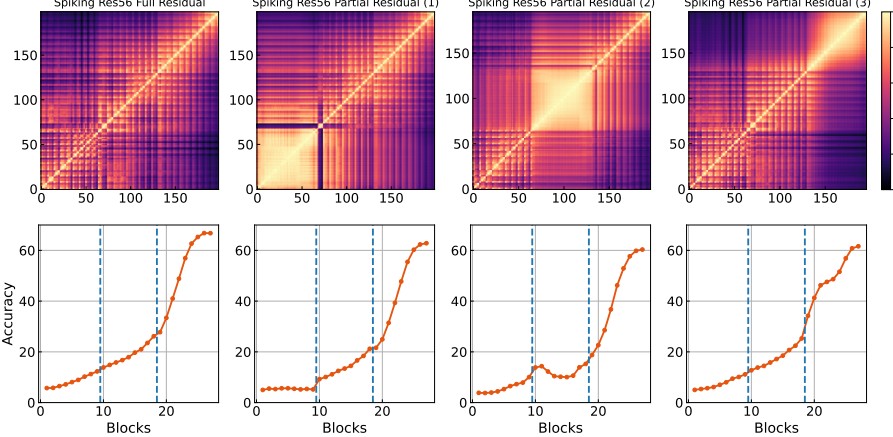

Figure A.6: **The effect of residual connections in the SNN.** We selectively disable residual connections in one of three stages in the ResNet-56. **Top**: the CKA heatmap of *SNN itself*, containing networks with different types of non-residual blocks. **Bottom**: The linear probing accuracy of each block.

the CKA heatmap and the CKA curve show little variations by changing the number of time steps. This confirms the results on CIFAR-10. In addition, we visualize the CKA similarities across time steps in each residual block. Fig. A.8 demonstrates that the first stage still produces temporal static features while the last stage has lower similarity across time steps.

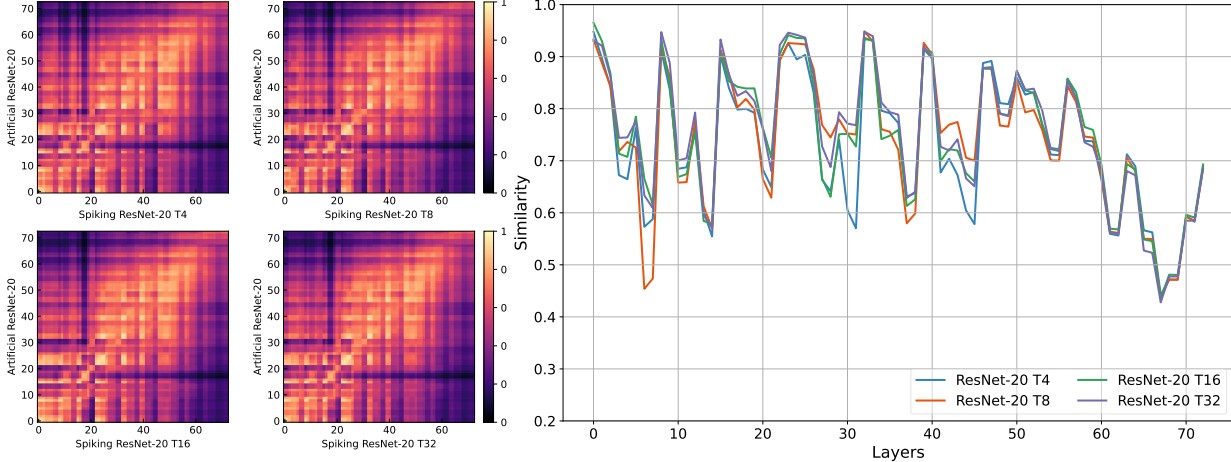

Figure A.7: **The effect of time steps in SNNs. Left**: CKA heatmaps between ANNs and SNNs with the different number of time steps. **Right**: The CKA curve of corresponding layers (diagonal values as in left).

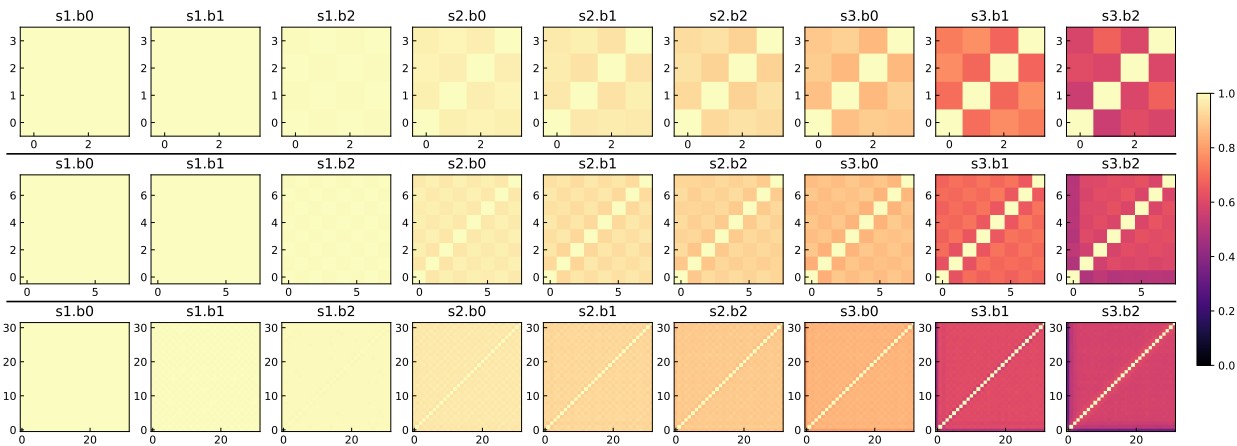

Figure A.8: **The similarity across times in SNN on CIFAR-100.** Each heatmap shows the CKA among different time steps in the output of residual block. "s" means stage, and "b" means block. The top/middle/bottom rows stand for spiking ResNet-20 with 4/8/32 time steps.

Finally, we rerun the adversarial robustness experiments on CIFAR-100. Here, we train two ResNet-20 with 4 times more channels and use PGD attack to measure the robustness against adversarial attack. Also, we plot the CKA curve between the feature of clean images and the feature of corrupted images. The results are shown in Fig. A.9.

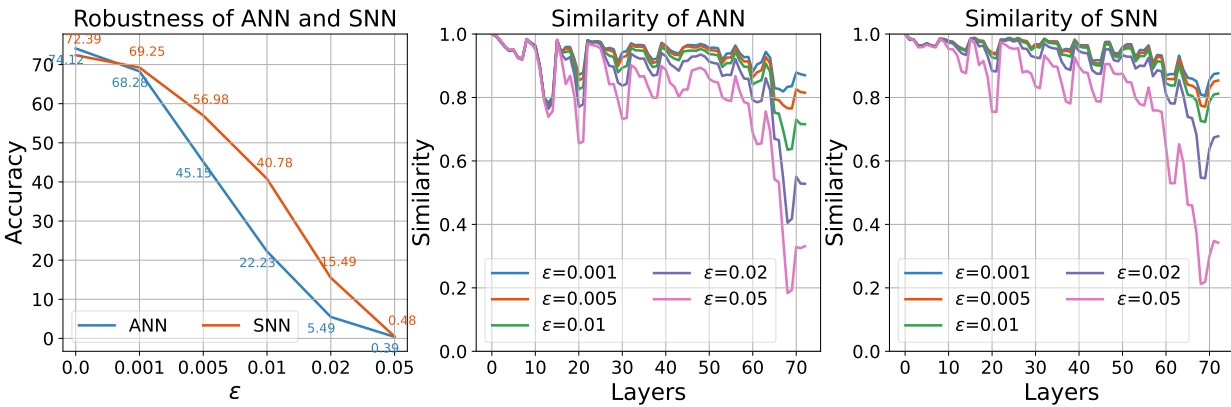

Figure A.9: **The robustness against adversarial attack on CIFAR-100. Left**: The accuracy of SNN and ANN after attack under different $\epsilon$. **Right**: The CKA curve between clean images and adversarial images of ANN and SNN, respectively.

### A.3 Results on CIFAR10-DVS

In this section, we conduct a representation similarity analysis on an event-based dataset—CIFAR10-DVS Li et al. (2017). As aforementioned in Fig. 7, the event dataset may generate different CKA heatmaps and curves when compared to the static dataset. Here, we first scale up the spatial dimensions in SNNs and ANNs for the CIFAR10-DVS dataset. As shown in Fig. A.10, we gradually increase either the depth to 110 layers or the width to 8 times as before.

Increasing the depth of ResNets on CIFAR10-DVS demonstrates a similar effect when compared to static CIFAR-10. Interestingly, the deep ResNet, for example, ResNet-110, emerges a similar periodical pattern to that on static CIFAR-10. However, we can find the peak CKA value is only around 0.75, (recall that the

peak CKA value on the CIFAR10 dataset is nearly 0.95, cf. Fig. 2), indicating the CIFAR10-DVS creates higher ANN-SNN difference in representations.

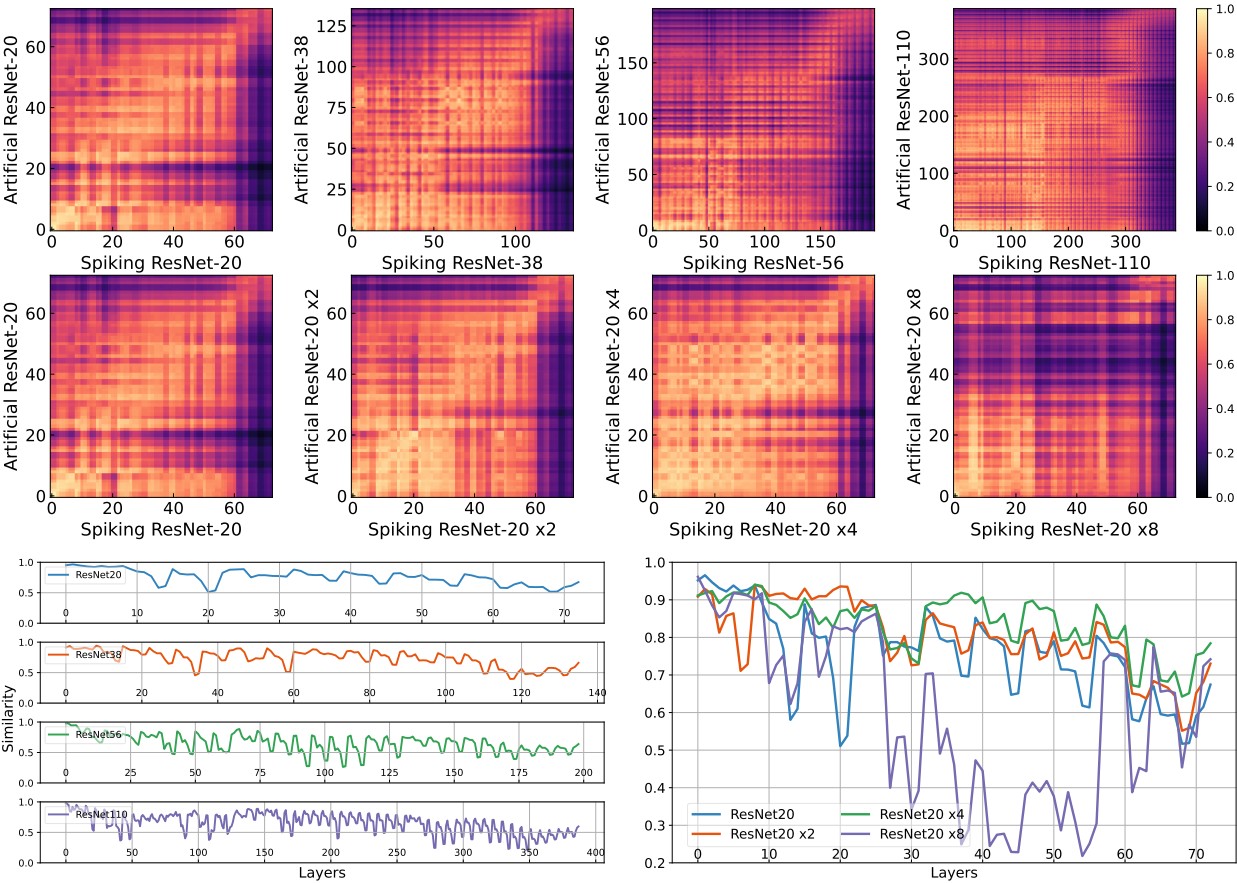

Figure A.10: **CKA heatmap between spiking and artificial ResNets with different depth and width on CIFAR10-DVS dataset. Top**: the CKA cross-layer heatmap across different depths from 20 layers to 110 layers. **Middle**: the CKA cross-layer heatmap across different widths from the original channel number to 8 times. **Bottom**: visualizing only the corresponding layer, which is the diagonal of the CKA heatmap.

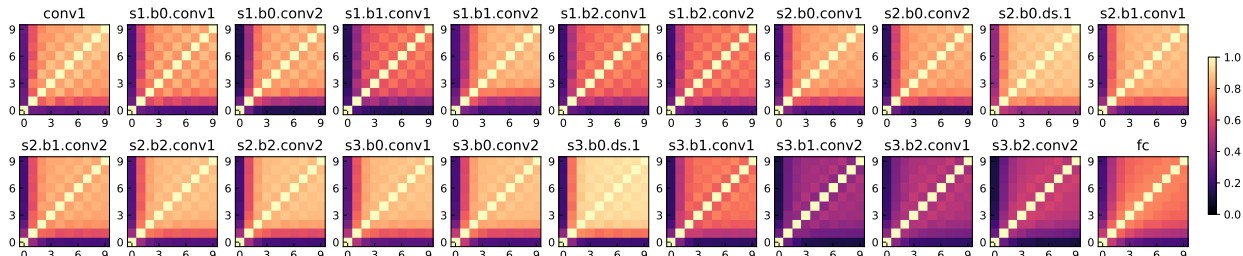

Figure A.11: **The similarity across times in SNN on CIFAR10-DVS.** Each heatmap shows the CKA among different time steps in the output of residual block. "s" means stage, and "b" means block. The model is trained with 10 time steps, i.e., 10 frames integrated for each CIFAR10-DVS data input.

For the extremely wide neural network, the trend holds for ResNet-20, ResNet-20 ($\times 2$), and ResNet-20 ($\times 4$). However, we find the ResNet-20 ($\times 8$) exhibits a significantly low CKA curve than other models. The difference may come from the artificial ResNet-20 ($\times 8$). According to the CKA heatmap, the $35$th$-60$th layers in artificial ResNet-20 ($\times 8$) become much darker than other heatmaps. We hypothesize that, with a

larger capacity of the neural network on the layer dynamics in the input data, the representation may change significantly.

We next study the internal CKA of each convolution in ResNet-20, which reveals the temporal dynamics of each convolution. As illustrated in Fig. A.11, the first convolution demonstrates dynamic features across 10 time steps, which shows different characteristics with static dataset (cf. Fig. A.12). Another notable difference is that the first and the second time step show very low similarity with other time steps. In summary, the rich temporal information dataset may increase the dynamics in SNNs across time steps and bring more differences when compared to ANNs.

### A.4   Similarity Across Time

In addition to the residual block, we also visualize the CKA heatmaps of convolutional layers and ReLU/LIF layers by comparing the similarity among different time steps. As can be seen in Fig. A.12, different from residual blocks, the similarity in convolutional and activation layers is more dynamic. Even in the first stage, the convolutional layers show different outputs across different time steps. This result further confirms our observations in residual blocks, where we found the convolutional and activation layers always decrease the ANN-SNN similarity while the residual block restores the similarity. Therefore, it suggests that a more temporal dynamic CKA heatmap may produce distinct features.

## B   Numerical Results

Here, we provide the clean accuracy of our trained models, both on CIFAR-10 and CIFAR-100. All models are trained with 300 epochs of stochastic gradient descent. The learning rate is set to 0.1 followed by a cosine annealing decay. The weight decay is set to 0.0001 for all models. The original ResNet-20 is a 3-stage model, each stage contains 2 residual blocks. The first stage contains 16 channels and the channels are doubled every time when entering the next stage. ResNet-38/56/110/164 contains 6/9/18/27 residual blocks in each stage. The wider networks just simply multiply all the channels by a fixed factor. We provide their top-1 accuracy in Table B.1.

Table B.1: The top-1 accuracy of SNNs and ANNs on CIFAR-10 (aka C10) and CIFAR-100 (aka C100) datasets.

| Layers | Width Factor | Time Steps | ANN (C10) | SNN (C10) | ANN (C100) | SNN (C100) |
|--------|--------------|------------|-----------|-----------|------------|------------|
| 20 | 1 | 4 | 91.06 | 89.67 | 64.23 | 61.51 |
| 38 | 1 | 4 | 92.34 | 91.14 | N/A | N/A |
| 56 | 1 | 4 | 92.98 | 91.94 | 68.39 | 66.63 |
| 110 | 1 | 4 | 92.37 | 91.83 | 69.20 | 66.95 |
| 164 | 1 | 4 | 93.00 | 92.05 | 70.27 | 67.09 |
| 20 | 2 | 4 | 93.36 | 92.00 | 70.14 | 68.65 |
| 20 | 4 | 4 | 94.52 | 93.96 | 74.12 | 72.39 |
| 20 | 8 | 4 | 94.78 | 94.48 | 76.57 | 75.81 |
| 20 | 16 | 4 | 94.78 | 94.73 | N/A | N/A |
| 20 | 1 | 8 | | 90.44 | | 63.03 |
| 20 | 1 | 16 | 91.06 | 90.98 | 64.23 | 64.34 |
| 20 | 1 | 32 | | 90.99 | | 64.33 |
| 20 | 1 | 64 | | 91.08 | | N/A |

## C   The Effect of Network Initialization

In this section, we study the effect of network initialization and verify if our findings are invariant to different initialization. Specifically, we train spiking and artificial ResNet-20s with two random seeds and measure the CKA similarity on these 4 models. Fig. C.1 shows the case of CKA heatmaps on the same model: the first two columns demonstrate the CKA heatmaps of the same initialization and the third column demonstrates the

CKA heatmaps of different initialization. We can find that SNNs or ANNs trained with different initialization share a high CKA similarity. Moreover, in Fig. C.2, we compare the CKA between SNNs and ANNs by permuting two different initialization. It can be seen that all four permutations have a similar distribution of CKA values. By comparing Fig. C.2 and Fig. C.1, we also find that the CKA heatmaps between ANNs and SNNs are also similar to the CKA heatmaps on the same model with two initializations. This indicates that the similarity between ANN and SNN is quite high.

## D  Network Architecture Details

Table D.1: The architecture details of ResNets.

|  | 20 layers | 38 layers | 56 layers | 110 layers | 164 layers |
|---|---|---|---|---|---|
| conv1 | $3 \times 3, 16, s1$ | $3 \times 3, 16, s1$ | $3 \times 3, 16, s1$ | $3 \times 3, 16, s1$ | $3 \times 3, 16, s1$ |
| block1 | $\begin{pmatrix} 3 \times 3, 16 \\ 3 \times 3, 16 \end{pmatrix} \times 3$ | $\begin{pmatrix} 3 \times 3, 16 \\ 3 \times 3, 16 \end{pmatrix} \times 6$ | $\begin{pmatrix} 3 \times 3, 16 \\ 3 \times 3, 16 \end{pmatrix} \times 9$ | $\begin{pmatrix} 3 \times 3, 16 \\ 3 \times 3, 16 \end{pmatrix} \times 18$ | $\begin{pmatrix} 3 \times 3, 16 \\ 3 \times 3, 16 \end{pmatrix} \times 27$ |
| block2 | $\begin{pmatrix} 3 \times 3, 32 \\ 3 \times 3, 32 \end{pmatrix} \times 3$ | $\begin{pmatrix} 3 \times 3, 32 \\ 3 \times 3, 32 \end{pmatrix} \times 6$ | $\begin{pmatrix} 3 \times 3, 32 \\ 3 \times 3, 32 \end{pmatrix} \times 9$ | $\begin{pmatrix} 3 \times 3, 32 \\ 3 \times 3, 32 \end{pmatrix} \times 18$ | $\begin{pmatrix} 3 \times 3, 32 \\ 3 \times 3, 32 \end{pmatrix} \times 27$ |
| block3 | $\begin{pmatrix} 3 \times 3, 64 \\ 3 \times 3, 64 \end{pmatrix} \times 3$ | $\begin{pmatrix} 3 \times 3, 64 \\ 3 \times 3, 64 \end{pmatrix} \times 6$ | $\begin{pmatrix} 3 \times 3, 64 \\ 3 \times 3, 64 \end{pmatrix} \times 9$ | $\begin{pmatrix} 3 \times 3, 64 \\ 3 \times 3, 64 \end{pmatrix} \times 18$ | $\begin{pmatrix} 3 \times 3, 64 \\ 3 \times 3, 64 \end{pmatrix} \times 27$ |
| pooling | Global average pooling | | | | |
| classifier | 10-d fully connected layer, softmax | | | | |

Table D.2: The architecture details of VGG networks.

|  | 13 layers | 19 layers | 25 layers | 31 layers | 43 layers |
|---|---|---|---|---|---|
| block1 | $(3 \times 3, 64) \times 1$ | $(3 \times 3, 64) \times 2$ | $(3 \times 3, 64) \times 2$ | $(3 \times 3, 64) \times 2$ | $(3 \times 3, 64) \times 2$ |
| pooling1 | Average pooling, $s2$ | | | | |
| block2 | $(3 \times 3, 128) \times 1$ | $(3 \times 3, 128) \times 2$ | $(3 \times 3, 128) \times 3$ | $(3 \times 3, 128) \times 4$ | $(3 \times 3, 128) \times 5$ |
| pooling2 | Average pooling, $s2$ | | | | |
| block3 | $(3 \times 3, 256) \times 2$ | $(3 \times 3, 256) \times 3$ | $(3 \times 3, 256) \times 4$ | $(3 \times 3, 256) \times 5$ | $(3 \times 3, 256) \times 6$ |
| pooling3 | Average pooling, $s2$ | | | | |
| block4 | $(3 \times 3, 512) \times 4$ | $(3 \times 3, 512) \times 8$ | $(3 \times 3, 512) \times 12$ | $(3 \times 3, 512) \times 16$ | $(3 \times 3, 512) \times 24$ |
| pooling | Global average pooling | | | | |
| classifier | 10-d fully connected layer, softmax | | | | |

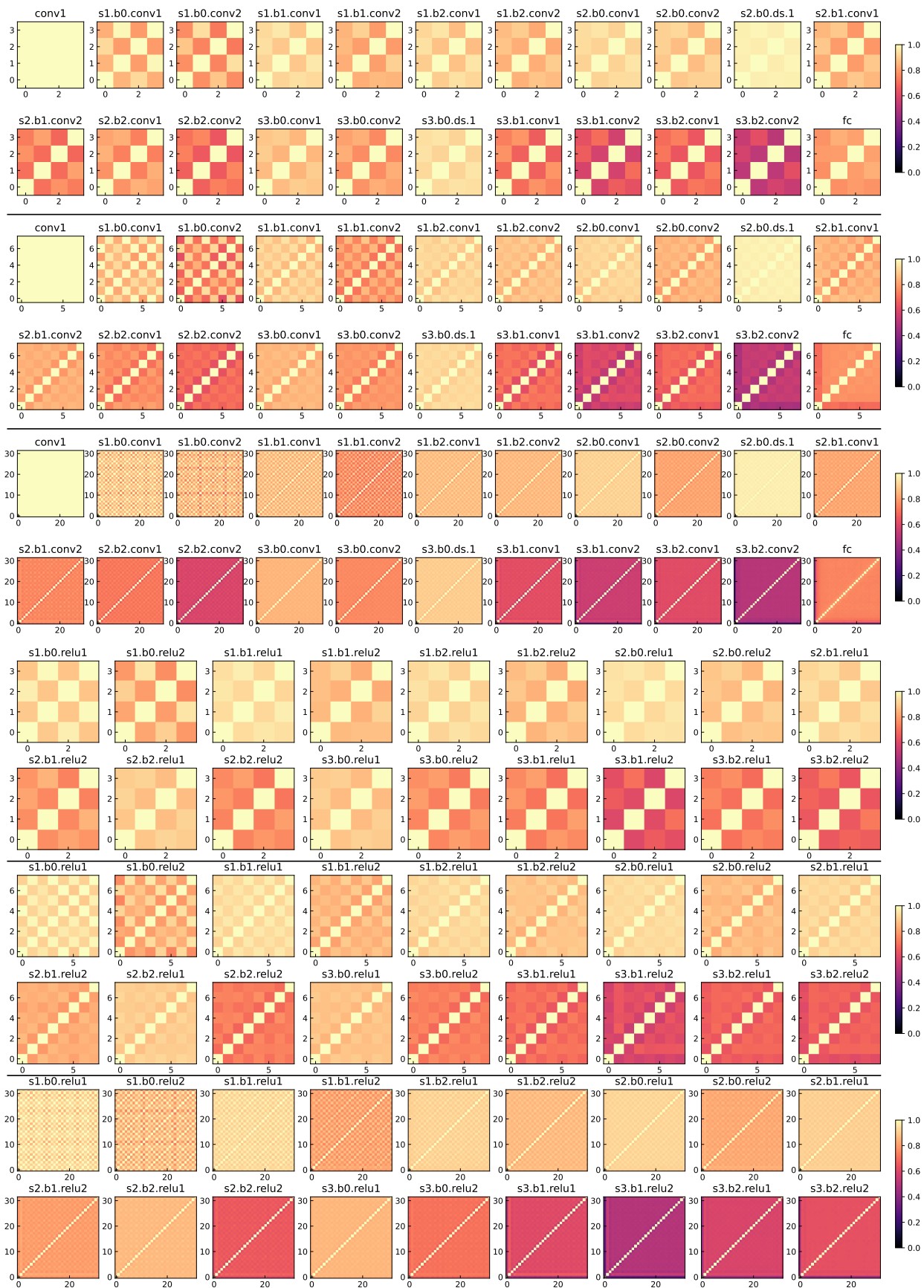

Figure A.12: **The effect of time steps in convolutional and activation layers of SNNs.**

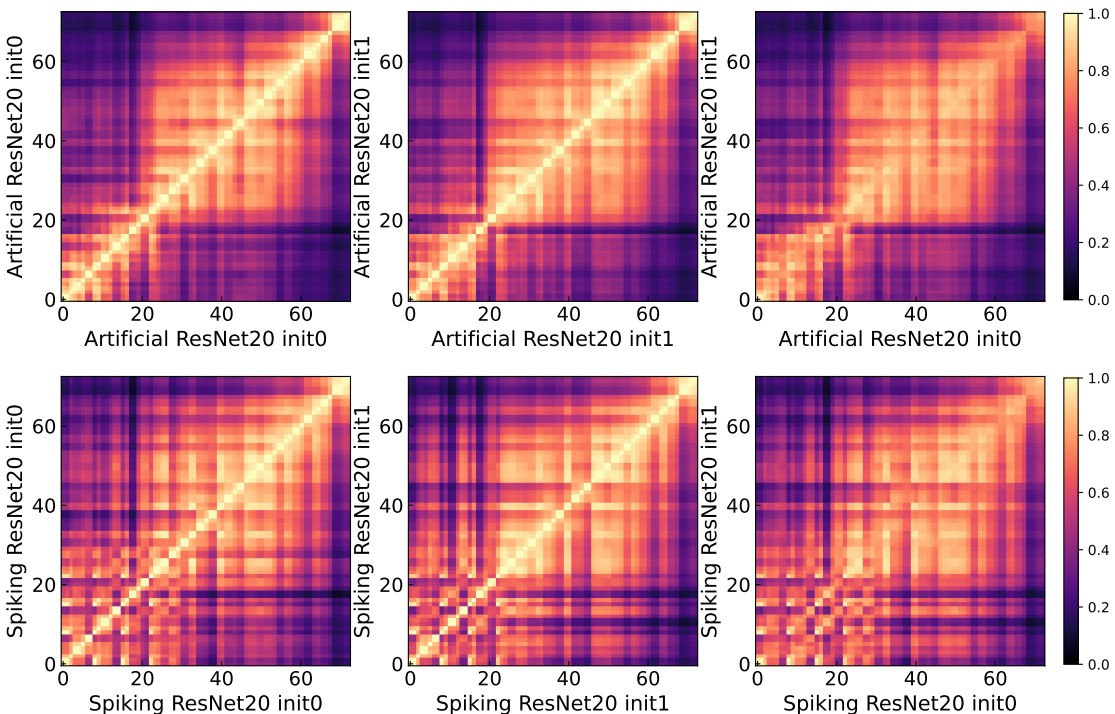

Figure C.1: **The similarity across different initialization on SNNs and ANNs.** We compare the CKA heatmap on the same model (optionally initialized with a different random seed).

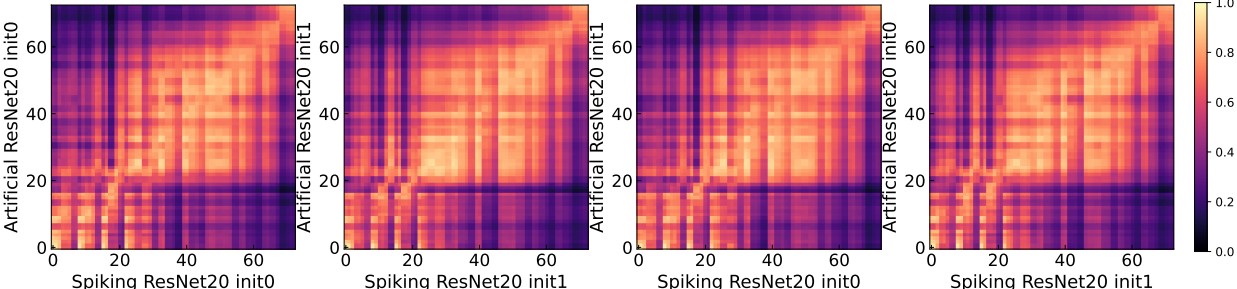

Figure C.2: **The similarity between ANN and SNNs across different initialization.**

