# OpenReview forum: "Uncovering the Representation of Spiking Neural Networks Trained with Surrogate Gradient"
_TMLR — Accepted by TMLR_

### Review · Reviewer_3q9t · 2023-01-20

**Summary Of Contributions:**

The paper describes a method to analyze the similarity between ANN and SNN where the definition of SNN is here restricted to an deep neural network (identifical to the correpsonding ANN) with replacement of the Relu activation with a "LIF-activation-function". The similarity is studied Central Kernal Alignment measure defined previously.

The authors find that
- within each block of a ResNet the network similarity is rather constant and fairly high. But between each residual connection, the difference rapidly increases before being brought back to high similarity right after each residual connection.
- In the lower layers the representation seems very time-invariant.
- The SNN seems to be a bit more robust to adversarial attacks.

**Audience:**

Yes

**Broader Impact Concerns:**

No concerns.

**Claims And Evidence:**

Yes

**Requested Changes:**

(1) In Section related work, please acknowledge the following sentence (or anything similar):
"there exists many other types of SNN: e.g. recurrent SNN or spike-timing SNN but this study focuses the type of feedforward SNN which has been shown to be equivalent to quantized ANN under a rate based hypothesis [ref 1 above]".

(2) It would be great to shown on similiraty matrix ANN Rest vs. another ANN ResNet of same size, this would provide a baseline for what is high/low similarity.

(3) In section of robustness to adversarial attacks or discussion, please acknowledge the following sentence (or anything similar):
"Quantized ANN are robust to adversarial attack, therefore it was expected that SNN which are roughly equivalent to quantized relu ANN [1] inherit this property.".

[2] Defensive Quantization: When Efficiency Meets Robustness
Ji Lin, Chuang Gan, Song Han

**Strengths And Weaknesses:**

It would be very interesting to study the representation differences between ANN and SNN but this definition of SNN here is a narrow view on what are spiking neural networks: here an SNN is simply a replacement of the ANN activation function with "LIF-activation" but there are many other implementation which would yield much more different representation (recurrent SNN, spike-time encoded SNN, etc...).

This also raises an important conceptual limitation because this type of LIF activation function are expected to yield a precise equivalence with quantized Relu network [1] under rate-coding hypothesis. Therefore using this type of conversion results, it means that we are basically studing here the quantization error or the difference in representation with an ANN and a quantized-relu ANN.

The last problem for me is that the work is not acknowledging that most the results would be identifical if studing a quantized relu. The robustness of quantized networks is infact well known. Is it not acceptable for SNN research to ignore (and not forget to cite) the ideas and progress from quantized ANN research.


[1] Optimal Conversion of Conventional Artificial Neural Networks to Spiking Neural Networks
Deng and Gu 2021

---

> ### Author Response · Authors · 2023-02-21
> **Reply to Reviewer 3q9t**
>
> Thank you for your detailed review and suggestions for our work. Below are our replies to your questions and we hope they address your concerns.
>
> Q1: “This definition of SNN here is a narrow view on what are spiking neural networks: here an SNN is simply a replacement of the ANN activation function with "LIF-activation" but there are many other implementation which would yield much more different representation (recurrent SNN, spike-time encoded SNN, etc...).”
>
> A1: We agree that the family of SNN is not restricted to the LIF neurons with surrogate gradient-based training algorithms and direct encoding of input. The reason we mainly focus on this type of SNN is that surrogate gradient training, in the current literature, achieves the highest accuracy with much smaller number of time steps. Other definitions of SNNs, like recurrent SNN and spike-time encoding SNN, use a lot more time steps and are hard to scale up to large datasets. [1] Therefore, we are interested in whether the current state-of-the-art SNNs are similar to ANNs or not.
>
> Q2: “This also raises an important conceptual limitation because this type of LIF activation function are expected to yield a precise equivalence with quantized Relu network [2] under rate-coding hypothesis.”
>
> A2: Thanks for your perspective observation. We partially agree with you that our studied type of LIF activation is similar to the quantized ReLU activation. However, we think these two are not precisely equivalent. The reasons are twofold:
>
> 1. In [2], the authors focus on the IF neuron while in our case we use the LIF which is associated with a leaky factor $\tau$ that is less than 1. Meanwhile, the IF neuron in [2] uses a soft-reset mechanism, which reduces the membrane potential by an amount of threshold after firing a spike. In our case, the reset is directly setting the membrane potential to 0, which is known as hard reset. These two differences make our LIF neuron no longer the same as the quantized ReLU activation.
>
> 2. Even if we use the IF neuron adopted in [2], there will be other conversion errors like unevenness error as suggested by [3], where the SNN may fire more or fewer spikes than the quantized value. Hence, we believe our SNNs still differ from the quantized ReLU ANNs.
>
> Q3: “The last problem for me is that the work is not acknowledging that most the results would be identifical if studing a quantized relu. The robustness of quantized networks is infact well known. Is it not acceptable for SNN research to ignore (and not forget to cite) the ideas and progress from quantized ANN research.
> In section of robustness to adversarial attacks or discussion, please acknowledge the following sentence (or anything similar): "Quantized ANN are robust to adversarial attack, therefore it was expected that SNN which are roughly equivalent to quantized relu ANN [1] inherit this property.".
>
> A3: Thanks for your suggestion. In our revision, we cite the work about defensive quantization [4] and acknowledge the robustness of this model. As we explained above, the SNN we used is still different from the quantized ReLU, therefore we revised it to: “Inspired from quantized ANNs that are robust to adversarial attack, the SNNs could also inherit this property since their activations are also discrete."
>
> Q4: It would be great to shown on similiraty matrix ANN Rest vs. another ANN ResNet of same size, this would provide a baseline for what is high/low similarity.
>
> A4: Thanks for your advice. We answer it in the general response and add this experiment in Appendix C. Please check them.
>
> References:
>
> [1] Wu, Yujie, et al. Direct training for spiking neural networks: Faster, larger, better. AAAI.
>
> [2] Deng, S., & Gu, S. (2021). Optimal conversion of conventional artificial neural networks to spiking neural networks. ICLR.
>
> [3] Bu, T., Fang, W., Ding, J., Dai, P., Yu, Z., & Huang, T. (2022). Optimal ANN-SNN conversion for high-accuracy and ultra-low-latency spiking neural networks. ICLR.
>
> [4] Lin, Ji, Chuang Gan, and Song Han. (2019) Defensive quantization: When efficiency meets robustness. ICLR.

---

### Review · Reviewer_91Rq · 2023-01-31

**Summary Of Contributions:**

This manuscript takes a careful look at the representational difference between spiking neural networks and artificial neural networks.  Through many different experiments, they identify that residual networks help maintain the predictive performance and representational similarity to artificial neural networks.  Conversely, the time dimension of spiking neural networks does not have a big effect.  The many different tests over many networks and detailed layer-level analysis adds to the broader set of literature exploring the relationships between spiking neural networks and artificial neural networks.

**Audience:**

Yes

**Broader Impact Concerns:**

N/A.

**Claims And Evidence:**

Yes

**Requested Changes:**

Major:
--
It is necessary to add description on how similarity should be interpreted, and discuss how dissimilar final layers in a neural network could be if those networks are predicting similar outcomes (see weakness above).  It is necessary to add more discussion and context to the discussion and conclusion under this viewpoint.

Please add a discussion on how much this framework could detect the scenario where the ANN and SNN follow different paths to a common final representation.

Minor:
--


> 3. The time dimension does not provide much additional representation power in SNNs. We also demonstrate that the shallow layers learn completely static representation along the time dimension. Even reducing the number of time steps to 1 in shallow layers does not significantly affect the performance of SNNs.

It does have a non-trivial impact in your own analysis, Table 2.  This is downplaying the effect too much.


> In the 3rd and 4th heatmaps, we can see that blocks without residual connections exhibit significantly different representations when compared to their former stage. Therefore, as long as ANN and SNN learn similar representations in the first layer, the similarity can propagate to very deep layers due to residual connections.

In Figure 4, I don't see how the visualizations support this conclusion.  Please add some clarifying comments and more detailed analysis.


> These results substantiate that the convolutional layers and LIF layers in SNNs are able to learn different representations than ANNs. However, the representation in the residual branch still dominates the representation in post-residual layers and leads to the junction of ANN’s and SNN’s representation.

This is an interesting claim.  Does this imply that the convolutional representations are much smaller than the residual?  At what point would this mean than an SNN follows a different path to a common representation with an ANN?


> This visualization is more specific and may accurately reveal the similarity between SNNs and ANNs because it only measures the CKA value of layers at the same positions.

I disagree with this claim as similarity could easily match to different layers if the networks are learned/defined differently.  If you want to keep this claim in there, please justify.


> Notably, we find the last 2/3 layers in artificial ResNet-164 exhibit significantly different representations than spiking ResNet-164, which demonstrates that deeper layers tend to learn disparate features.

I cannot visually discern this from the figures.  I would suggest quantifying for clarity.


I do not see a definition for PGD.


>Figure 6:...The top/middle/bottom rows stand for spiking ResNet-20 with 4/8/32 time steps.

This should state they are those networks, not stand for.


There are several grammar errors, please proofread again.

**Strengths And Weaknesses:**

Strengths:
--
The biggest strength of this manuscript is the detailed experimentation, including how several different changes to spiking and artificial neural networks (SNN/ANN) impact their relational similarity.  This is also performed over many different networks and has detailed information on how blocks and residual connections impact similarity in these neural networks.  This detailed analysis can help add to the understanding of how these networks are trained.

Weaknesses:
--
The biggest weakness is the lack of discussion on what the representational similarity really means, and how dissimilar two networks can become.  For example, if a SNN and an ANN both have accurate predictions, this all but necessitates that the representation at the final layer is quite high.  As such, I would expect representational similarity to stay high throughout the neural networks.  If the final layers were dissimilar, it seems like there would be vastly different performance.  Thus, I struggle with how to interpret this final claim:
> Unfortunately, our results may not fully support the opinion that SNNs learn effective and distinct spatial- temporal representation compared to the spatial representation in ANNs. Current SNN learning relies on the residual connection and wider neural networks (for example, Zheng et al. (2020) use ResNet-19 which is similar to our ResNet-20 8×) to obtain decent task performance. However, our study suggests that this task performance is highly credited to the similar representation with ANN.

I would encourage the authors to discuss how representational similarity could be different.  For example, if the blocks are dissimilar in SNN and ANN, does that mean that the paths to the final representation are different even if the final representation is the same?


Additionally, several claims are a bit stronger than the empirical evidence, which can be handled with mild language edits (see requested changes).

---

> ### Author Response · Authors · 2023-02-21
> **Reply to Reviewer 91Rq (1/2)**
>
> Thank you for your detailed comments and suggestions. Below are our detailed replies to each question.
>
> Q1: “The biggest weakness is the lack of discussion on what the representational similarity really means, and how dissimilar two networks can become. For example, if a SNN and an ANN both have accurate predictions, this all but necessitates that the representation at the final layer is quite high. As such, I would expect representational similarity to stay high throughout the neural networks. If the final layers were dissimilar, it seems like there would be vastly different performance”
>
> A1: Thank you for raising this concern and for the suggestions. We’d like to clarify that the representation similarity does not necessarily correlate to the accuracy difference between the two models. The CKA metric we use measures the difference between covariance matrices across the batch dimension in ANN/SNN. Basically, the CKA checks whether two models treat different inputs in the same way. Therefore, even if the accuracy between ANN and SNN is similar, the representation can be different depending on how they handle each class. In an extreme example, consider a binary classification problem where model A always predicts the first class and model B always predicts the second class, their accuracies are both 50% (assuming the two classes have the same amount of input), but their CKA value would be 0 because their activations are completely different.
>
> Q2: I struggle with how to interpret this final claim.
>
> A2: We provided a discussion on the baseline of the CKA metric in the General Response. Here, we’d like to clarify our final claim. Our final claim is based on the fact that the CKA between ANN/SNN is higher than expected. Normally, researchers characterize SNNs as third-generation neural networks due to their biological plausibility. However, based on our analysis, we find that SNNs are not so different from ANNs, especially when the channel number is high (>0.8 CKA similarity in almost every corresponding layer). This observation means that the wider SNN adopted in the current literature [2] only brings a similar representation to ANN rather than leveraging the unique feature extractions. To better support this claim, we also add an accuracy table in Section 4.1.
>
>
> Q3: I would encourage the authors to discuss how representational similarity could be different. For example, if the blocks are dissimilar in SNN and ANN, does that mean that the paths to the final representation are different even if the final representation is the same?
>
> A3: Essentially, the representation similarity metric measures how the neural network handles different input images. To correlate the representation similarity metric with accuracy is not straightforward. For example, [1] discusses the CKA between vision transformer and convnets, where they find the CKA values are very low (<0.4), but these two models have a similar accuracy level. Again, we refer you to our general response for how representation similarity could be different. As for the example, we think that if the blocks are dissimilar, the final representation is unlikely to be similar in the end.
>
> References:
>
> [1] Raghu, M., Unterthiner, T., Kornblith, S., Zhang, C., & Dosovitskiy, A. (2021). Do vision transformers see like convolutional neural networks? NeurIPS
>
> [2] Zheng, H., Wu, Y., Deng, L., Hu, Y., & Li, G. (2021). Going deeper with directly-trained larger spiking neural networks. AAAI

---

> ### Author Response · Authors · 2023-02-21
> **Reply to Reviewer 91Rq (2/2)**
>
> **Minor:**
>
> Q1: It does have a non-trivial impact in your own analysis, Table 2. This is downplaying the effect too much.
>
> A1: Thanks for the question, we think Fig 6 or Table 2 does not downplay our analysis. Table 2 indicates that the time dimension indeed does not have much influence on shallow layers since reducing the time step to 1 in shallow layers brings a marginal accuracy drop, thereby verifying our CKA results in Fig 6.
>
> Q2: In Figure 4, I don't see how the visualizations support this conclusion. Please add some clarifying comments and more detailed analysis.
>
> A2: Thank you for the suggestion. Here, the reasons are two-fold. First, from the 3rd and 4th heatmaps, we can find that the stage without residual connections has low CKA values (<0.4) compared to the early stages. Second, the stage with residual connections, on the contrary, can have high CKA values than the early stages. These two factors imply residual connections can preserve the representation in previous layers. We also revise the context in the paper to make it clearer.
>
> Q3: This is an interesting claim. Does this imply that the convolutional representations are much smaller than the residual? At what point would this mean than an SNN follows a different path to a common representation with an ANN?
>
> A3: Yes, we also think the convolutional representations are smaller than the residual so that each time after the element-wise addition the CKA value restores to above 0.9. At this point of time, we could not determine an accurate threshold to say if SNN differs from ANNs in terms of representation. But a 0.9 CKA value is indeed high to say SNN and ANN are much more similar than we expected. Please also refer to the general response for the baseline of CKA values.
>
> Q4: I disagree with this claim as similarity could easily match to different layers if the networks are learned/defined differently. If you want to keep this claim in there, please justify.
>
> A4: Sorry for missing some explanation here. Since SNNs and ANNs share the same architecture, with only the activation and normalization layers being different, checking the corresponding layer’s representation can give a more accurate measure. If the network topology is different, for example, spiking ResNet-20 vs. artificial ResNet-56, then measuring the corresponding layer has no meaning.
>
> Q5: I cannot visually discern this from the figures. I would suggest quantifying for clarity.
>
> A5: As can be seen in the top right corner of Fig. 2, the artificial ResNet-164 has a large area of purple (corresponding to <0.3 CKA values). We revise this sentence to add a more quantitative description.
> [add sentence in text]
>
> Q6: I do not see a definition for PGD.
>
> A6: Sorry for the missing definition of PGD, we’ve added it now.
>
> Q7: This should state they are those networks, not stand for; There are several grammar errors, please proofread again.
>
> A7: Thank you for these corrections, we go over one more proofreading again.

---

### Review · Reviewer_LSrQ · 2023-02-06

**Summary Of Contributions:**

This paper presents a similarity analysis between spiking neural networks (ANN) trained with surrogate gradient method and artificial neural networks (ANN). Using extensive analysis with different versions of the Resnet network, the authors show that Residual connections improve similarity of SNN and ANNs.

Additionally, they show that temporal dimension in SNN is not likely necessary of image classification tasks, the representation is likely very different based on the task and SNNs are more robust to adversarial attacks than ANNs.

**Audience:**

Yes

**Broader Impact Concerns:**

No ethical concerns.

**Claims And Evidence:**

Yes

**Requested Changes:**

• Add more details on the accuracy of SNN, how it was evaluated, and relative performance of each architecture with respect to ANN.

• Add a figure showing the correlation between accuracy of SNN and similarity of SNN to ANN. If there is the tight correspondence, it would
be help the main point of the paper.

• Similarity curves for each block in Figure 3 could be plotted on top of each other, and averaged to see if effect of different layers is consistent in each block. For example, the second LIF stage gives a big drop in similarity in 34th residual block, whereas both the LIF stages give similar drop in 10th residual block. Averaging across multiple blocks might be insightful.

* All curves and similarity matrices should be averaged across multiple networks, trained with different initializations. Or, show some evidence of consistency across initializations.


**Strengths And Weaknesses:**

The paper is written well, though the language/phrasing can be made clearer in some regions. The analyses are described sufficiently in detail. Analyses seem technically correct.

One of the main weakness of the paper is that the analysis is limited to Resnet, and image classificiation task. With event-based task, the results are very different. This reduces my confidence generality of the results.

Axes labels missing in some places (ex. Fig 4).

---

> ### Author Response · Authors · 2023-02-21
> **Reply to Reviewer LSrQ**
>
> Thank you for the thoughtful review.  Below are our replies to your questions and we hope they address your concerns.
>
> Q1: One of the main weaknesses of the paper is that the analysis is limited to Resnet, and image classification task. With event-based task, the results are very different. This reduces my confidence generality of the results.
>
> A1: Thanks for the question. In Appendix A we provide CKA results on VGG networks and CIFAR-100 datasets. Admittedly, the CKA distribution on the event-based dataset is different from the static image dataset. However, we think this phenomenon is not surprising. The SNNs inherently fit more on the event-based dataset since they can accept input with a time dimension, while ANNs have to infer the averaged sample. Given that we find SNNs are more similar to ANNs than expected on the static datasets, studying SNNs on the event-based dataset may be more valuable in the future.
>
> Q2: Axes labels missing in some places (ex. Fig 4).
>
> A2: Thank you for your advice. We provide the axes now.
>
> Q3: Add more details on the accuracy of SNN, how it was evaluated, and relative performance of each architecture with respect to ANN.
>
> A3: Thanks for this advice. Now we add the accuracy results in Table 1 (Section 4.1), which shows that wider networks tend to have a smaller accuracy gap while deeper networks do not significantly mitigate the accuracy gap.
>
> Q4: Similarity curves for each block in Figure 3 could be plotted on top of each other, and averaged to see if effect of different layers is consistent in each block. For example, the second LIF stage gives a big drop in similarity in 34th residual block, whereas both the LIF stages give similar drop in 10th residual block. Averaging across multiple blocks might be insightful.
>
> A4: Thanks for this useful suggestion, we now average the CKA values from all residual blocks and plot them in Fig 3 top right corner. The drop in both LIF layers is similar on average.
>
> Q5: All curves and similarity matrices should be averaged across multiple networks, trained with different initializations. Or, show some evidence of consistency across initializations.
>
> A6: Thanks for your advice, we agree that network initialization is a key factor in representation similarity analysis. Due to the limited time in this rebuttal, we could not re-run all experiments with multiple initializations, therefore we run initialization comparison experiments in Appendix C now. The results show that different initializations cause consistent CKA curves on ResNet-20. In the future version, we will add more initialization experiments.

---

### Author Response · Authors · 2023-02-21
**General Response**


We would like to thank all the reviewers for their constructive feedback. We are delighted to see that reviewers acknowledge our efforts in conducting the representation similarity analysis between ANNs and SNNs. We will reply to your comments one by one, but in the beginning, we'd like to have a general response, clarifying a common question. We also change our manuscript highlighted with blue color.

All three reviewers asked about the baseline of the CKA measure we use. Specifically, Reviewer LSrQ is concerned about the impact of network initialization, Reviewer 91Rq is concerned about how CKA values can be different in two networks, and Reviewer 3q9t asked for a comparison of two same types of neural networks to see what is high/low in terms of CKA.

We thank the reviewers for raising this good question. First,  we have conducted a study in Appendix C regarding networks trained with different initializations where we use two random seeds to train spiking and artificial ResNet-20. Notably, we find the CKA heatmaps are consistent across two initializations. Moreover, the CKA heatmaps between ANNs and SNNs are very similar to the CKA heatmaps between two ANNs trained with different random seeds. This finding confirms that the similarity between SNNs and ANNs is high as it produces similar CKA heatmaps compared to the heatmaps of two same models.
Second, we also quote the conclusion from another paper. According to paper [1], it is discovered that ViT and CNNs have dissimilar representations where their CKA heatmaps have a large purple area (<0.4 CKA). Presumably, if SNNs learn different representations than ANNs we would expect the CKA heatmap should have a lot more purple area. However, we find the heatmap remains bright and the corresponding layers have high CKA values (>0.8). Therefore, in this paper, we claim that SNNs and ANNs are indeed more similar than we expected.

References:

[1] Raghu, M., Unterthiner, T., Kornblith, S., Zhang, C., & Dosovitskiy, A. (2021). Do vision transformers see like convolutional neural networks? NeurIPS

---

### Comment · Action_Editors · 2023-03-28
**One more revision requsted before finalized decision**

Given my initial notes in preparing a decision, the Editors In Chief suggested that there be one more round of revision on this paper before a decision is finalized.  The authors should aim to address these concerns in roughly 2 weeks if possible.

---

> ### Comment · Action_Editors · 2023-03-28
> **Notes from the editor**
>
> The overall paper is sound.  In my opinion, the scholarship (contextualization relative to previous work) is mostly quite clear and thorough.  The work that was done was well prepared and presented, with clear figures and discussion of the findings.  That said, I would encourage one more round of proof-reading for minor grammar issues.
>
> In general, the authors made a sincere attempt to address reviewer comments.
>
> Reviewer LSrQ was positive about the technical correctness of the paper and made useful suggestions, which the authors incorporated into the paper, including the presentation of model performance (table 1).
>
> Reviewer 91Rq encouraged the authors to discuss how representational similarity could be different when there is similar performance. The authors discuss what they propose to be a simple illustrative example as a response, but I'm not certain I believe it is a good example (if two representations are perfectly anti-correlated, wouldn't they have high representational similarity?).
> - It would be better to find and reference a research paper that shows varying levels of similarity for different networks with similar task performance.
>
> Reviewer 3q9t raised a point that the authors of this paper use a quite restrictive definition of an SNN: "SNN here is a narrow view on what are spiking neural networks: here an SNN is simply a replacement of the ANN activation function with "LIF-activation" but there are many other implementation which would yield much more different representation (recurrent SNN, spike-time encoded SNN, etc...)." and expands "This also raises an important conceptual limitation because this type of LIF activation function are expected to yield a precise equivalence with quantized Relu network [1] under rate-coding hypothesis. Therefore using this type of conversion results, it means that we are basically studying here the quantization error or the difference in representation with an ANN and a quantized-relu ANN." While the authors reply to the reviewer that the SNN model they consider uses LIF and is not exactly quantized, the reviewer's broader point is basically that we should view the setting considered by the authors as at least approximately similar.  As such, our expectation is that insofar as the SNNs being considered here might be seen as approximations of the ANNs (with performance that is usually slightly worse), these SNNs may be expected to have rather similar representations compared to the ANNs.
>
> So, contrary to the surprise expressed by the authors, it seems somewhat intuitive to me that SNNs that are specifically trained by surrogate gradients may be similar to ANNs trained by backprop.  SNNs generally are a more broad class and many SNNs perform inference using engineering rules or coding schemes.  As such:
> - It wouldn't hurt for the authors to be more explicit that training SNNs by surrogate gradients isn't the only approach to producing SNNs.
> - Clarify the statement: "Unfortunately, our results may not fully support the opinion that SNNs learn effective and distinct spatial-temporal representation compared to the spatial representation in ANNs".  Honestly, it was not obvious until here that you were "hoping" to see distinct representations.
>
> Nevertheless, I agree that it is valuable and interesting to confirm that this particular learning approach results in SNNs that are similar to ANNs.
>
> In addition to the points above, from my own reading, there are a few overstatements that stand out:
>
> - "Recently, researchers have demonstrated that SNNs are able to achieve state-of-the-art performance in image recognition tasks using surrogate gradient training." -- I believe this refers to results achieving "nearly" state-of-the-art results.  If I understand correctly, SNNs aren't better than standard ANNs, but have recently been able to be trained to *nearly* the same performance using surrogate gradients.
> - "With this method, SNNs can be optimized by Backpropagation Through Time (BPTT) (Werbos, 1990) algorithm, delivering state-the-ofthe-art task performance." -- It seems odd to be treating the Werbos reference alone as the core ingredient, since the architectural choices matter a lot for performance in addition to the learning algorithm.  (Also typo in "state-of-the-art" here.)
> - "SNNs are recognized as a candidate for next-generation artificial intelligence." -- Roughly this statement appears in a few places (opening of the abstract and opening of section 2).  In the abstract, I think it is fairly phrased, largely because the energy efficiency statement is made and it says "next-generation neural networks".  In the section 2 variant of this statement, I think is is a bit bold ("next-generation artificial intelligence") and could be misread as implying that SNNs are somehow more "intelligent"; however, this is not true given the current state of the field.
>
> Please address these issues (discussed in bullets above), and then we'll finalize.

---

> > ### Author Response · Authors · 2023-04-11
> > **Response to Notes from Editor**
> >
> > We’d like to thank the editor and reviewers for their thoughtful comments and suggestions. We have revised our paper according to the suggestions from reviewers. All updates are highlighted with orange colors.
> >
> > **Reply to Reviewer 91Rq**:
> >
> > We want to clarify our example in our previous response first. “if two representations are perfectly anti-correlated, wouldn't they have high representational similarity?". Here, if two networks are perfectly anti-correlated, their CKA value would become 0, meaning the lowest representation similarity. It is still possible that they can have the same accuracy (50% vs 50%).
> >
> > Regarding *a research paper that shows varying levels of similarity for different networks with similar task performance*, we find Fig. 6 in [1] shows CKA heatmaps between networks initialized with different random seeds, where the heatmaps show a variety of different CKA values. Ideally, two same network architectures with different initializations should have very similar representations. However, [1] finds that CKA value can become extremely low for very deep and wide networks.
> >
> > [1] Nguyen T, Raghu M, Kornblith S. Do wide and deep networks learn the same things? uncovering how neural network representations vary with width and depth[J]. arXiv preprint arXiv:2010.15327, 2020.
> >
> > **Reply to Reviewer 3q9t**:
> >
> > Suggestion 1: It wouldn't hurt for the authors to be more explicit that training SNNs by surrogate gradients isn't the only approach to producing SNNs.
> >
> > Answer: Thanks for your suggestion. In our introduction, we change our description to “various training techniques have been proposed. For example, spike-timing-dependent plasticity (STDP) (Rao & Sejnowski, 2001) either strengthens or weakens the synaptic weight based on the firing time; time-to-first-spike (Mostafa, 2017) encodes the information into the time of spike arrival to get a closed-form solution of the gradients. However, these two methods are restricted to small-scale tasks and datasets. Surrogate gradient technique (Bengio et al., 2013; Bender et al., 2018; Wu et al., 2018; Bellec et al., 2018), on the other hand, can achieve the best task performance by applying an alternate function during back-propagation.”
> >
> > Suggestion 2: Clarify the statement: "Unfortunately, our results may not fully support the opinion that SNNs learn effective and distinct spatial-temporal representation compared to the spatial representation in ANNs"
> >
> > Answer: We change it to a more neutral tone: Our results show that SNNs optimized by surrogate gradient algorithm do not learn distinct spatial-temporal representation compared to the spatial representation in ANNs.
> >
> > In addition, in the next paragraph, we mentioned surrogate gradient methods are inherently bio-implausible and resemble the optimization method for ANNs.
> >
> > Suggestion 3: If I understand correctly, SNNs aren't better than standard ANNs, but have recently been able to be trained to nearly the same performance using surrogate gradients.
> >
> > Answer: Thank you for this correction. We agree to add *nearly* is much better.
> >
> > Suggestion 4: It seems odd to be treating the Werbos reference alone as the core ingredient since the architectural choices matter a lot for performance in addition to the learning algorithm. (Also typo in "state-of-the-art" here.)
> >
> > Answer: Originally, we wanted to emphasize that the surrogate gradient achieves the best performance compared to other learning algorithms. To reduce ambiguity, we change our reference and description to “Combined with surrogate gradient, SNNs can be optimized by Backpropagation Through Time (BPTT) (Neftci et al., 2019) algorithm, outperforming other learning rules in SNNs”.
> >
> > Suggestion 5: "SNNs are recognized as a candidate for next-generation artificial intelligence." -- Roughly this statement appears in a few places (opening of the abstract and opening of section 2). In the abstract, I think it is fairly phrased, largely because the energy efficiency statement is made and it says "next-generation neural networks". In the section 2 variant of this statement, I think is is a bit bold ("next-generation artificial intelligence") and could be misread as implying that SNNs are somehow more "intelligent"; however, this is not true given the current state of the field.
> >
> > Answer: Thank you for this advice. We agree with your suggestion and change it to “SNNs have gained increasing attention for building low-power intelligence.” We also finished proofreading for some grammatical errors.

---

### Decision · Action_Editors · 2023-03-24

**Recommendation:** Accept as is

**Comment:**

This paper had an additional round of revision before the decision was reached.  The authors addressed all concerns that were raised.  I now endorse it to be accepted as is.

**Audience:**

I am reasonably confident that the findings would be of interest to a subset of the TMLR audience.

**Claims And Evidence:**

This paper analyzes the similarity of representations of standard artificial neural networks (ANNs) for image recognition versus the representations that arise in similarly architected spiking neural networks (SNNs) as a result of surrogate gradient training.

The core claims of the paper seem reasonable, and the claims are supported by seemingly technically sound experiments and analysis.